# Deregulated DNA ADP-ribosylation impairs telomere replication

Anne R. Wondisford[1], Junyeop Lee [2,6], Robert Lu[3,6], Marion Schuller[4],
Josephine Groslambert [4], Ragini Bhargava[1], Sandra Schamus-Haynes[1],
Leyneir C. Cespedes[2], Patricia L. Opresko [1,5], Hilda A. Pickett [3],
Jaewon Min [2], Ivan Ahel [4] & Roderick J. O'Sullivan [1] ✉

The recognition that DNA can be ADP ribosylated provides an unexpected regulatory level of how ADP-ribosylation contributes to genome stability, epigenetics and immunity. Yet, it remains unknown whether DNA ADP-ribosylation (DNA-ADPr) promotes genome stability and how it is regulated. Here, we show that telomeres are subject to DNA-ADPr catalyzed by PARP1 and removed by TARG1. Mechanistically, we show that DNA-ADPr is coupled to lagging telomere DNA strand synthesis, forming at single-stranded DNA present at unligated Okazaki fragments and on the 3' single-stranded telomere overhang. Persistent DNA-linked ADPr, due to TARG1 deficiency, eventually leads to telomere shortening. Furthermore, using the bacterial DNA ADP-ribosyl-transferase toxin to modify DNA at telomeres directly, we demonstrate that unhydrolyzed DNA-linked ADP-ribose compromises telomere replication and telomere integrity. Thus, by identifying telomeres as chromosomal targets of PARP1 and TARG1-regulated DNA-ADPr, whose deregulation compromises telomere replication and integrity, our study highlights and establishes the critical importance of controlling DNA-ADPr turnover for sustained genome stability.

ADP-ribosylation (ADPr) is a modification of macromolecules catalyzed by ADP-ribosyl-transferase (ART) enzymes known as PARPs that has a vital role in cellular physiology, but most pertinently in safeguarding genome stability through DNA repair[1]. PARP activity is counteracted by ADP-ribosyl-hydrolases, including PARG (poly-ADP-ribose glycohydrolase), the major enzyme that degrades PAR chains and other mono-ADPr specific enzymes such as ARH3 (ADP-ribosyl-hydrolase 3)[1]. Although ADPr has long been considered solely a posttranslational modification of proteins, recent evidence for the ADP-ribosylation of nucleic acids, including DNA, challenges the established views of this modification[2–5]. The 5' and/or 3' phosphorylated exposed ends were determined

to be substrates for in vitro DNA ADP-ribosylation (DNA-ADPr) by PARPs 1–3 (refs. 3,6,7). Bacterial DNA ADP-ribosyl-transferase (DarT) toxins, DarT1 and DarT2, were identified as ancestral PARP1-like ADP-ribosyltransferases that catalyze the sequence-specific ADPr of guanosine[8] and thymidine[9] bases in single-stranded DNA (ssDNA), respectively. In bacteria, thymidine-linked ADP-ribose is reversed by the DNA ADP-ribosyl glycohydrolase (DarG) antitoxin. Modeling of human TARG1 (terminal ADP-ribosyl glycohydrolase 1), an ADP-ribosyl-hydrolase known to reverse glutamate and aspartate-linked protein-ADPr[10,11], revealed structural conservation between its catalytic macrodomain and that of DarG[11,12]. Furthermore, TARG1 and DarG

[1]Department of Pharmacology and Chemical Biology, UPMC Hillman Cancer Center, University of Pittsburgh, Pittsburgh, PA, USA. [2]Institute for Cancer Genetics, Columbia University Vagelos College of Physicians and Surgeons, New York, NY, USA. [3]Telomere Length Regulation Unit, Children's Medical Research Institute, Faculty of Medicine and Health, University of Sydney, Westmead, New South Wales, Australia. [4]Sir William Dunn School of Pathology, University of Oxford, Oxford, UK. [5]Department of Environmental and Occupational Health, University of Pittsburgh, Pittsburgh, PA, USA. [6]These authors contributed equally: Junyeop Lee, Robert Lu. ✉e-mail: osullivanr@upmc.edu

can functionally complement each other in reversing DarT-induced DNA-ADPr in human and bacterial cells[12,13]. Thus, the pathways that regulate DNA-ADPr may be conserved from bacteria to humans. Recently, rare PARP1-mediated DNA-ADPr of adenine bases was identified in mammalian cells[14]. However, where DNA-ADPr occurs in the genome, how it contributes to genome stability and whether TARG1 controls it remains unknown. Addressing these questions is paramount for advancing our understanding of this emerging modification.

## Results

### Regulation of ADPr of telomeric DNA by PARP1 and TARG1

Telomeres are essential for genome integrity. PARP1 is recruited to telomeres to repair internal telomere DNA breaks and base lesions[15,16] and can promote telomere fusions via the alternative end-joining mechanism[17]. Furthermore, telomeric DNA terminates with a single-stranded 3′ overhang and recessed 5′ end, a potent trigger of PARP1 activity, that is shielded by the telomere-binding and protection complex, Shelterin[18]. In assessing patterns of nuclear ADP-ribosylation by immunofluorescence, we observed a pronounced accumulation of PAR foci in TARG1-deficient cells that colocalized with telomeres, marked by the telomere-binding protein, TRF1 (Extended Data Fig. 1a). Considering the strong evidence for PARP1 dependent activities at telomeres, as well as the putative link of TARG1 in reversing DNA-ADPr, this prompted us to investigate whether the modification occurs on telomeric DNA in human cells.

The chemical properties of the ADP-ribose linkage in DNA have precluded its detection and genomic assignment by conventional immunoprecipitation-PCR or next-generation sequencing methods[14]. We used a region-specific extraction (RSE) methodology that relies on the specific hybridization of a biotin-conjugated (AATCCC) oligonucleotide to the TTAGGG-rich telomere overhang followed by streptavidin pulldown from purified genomic DNA (Fig. 1a)[19]. The enrichment of telomeric DNA repeat sequences by this methodology and preservation of ADP-ribose due to the absence of TARG1 hydrolytic activity could enable the detection of DNA-linked ADP-ribose in dot blot using specific anti-ADP-ribose antibodies. The enrichment of telomeric DNA from control and TARG1-deficient U2OS cells was verified in Southern blot with radiolabeled telomere-specific probes (Fig. 1a). In contrast, Alu repeat sequences were not enriched in the telomere RSE samples (Fig. 1a). Western blotting using specific antibodies confirmed the equal capture of double-stranded (dsDNA) while revealing greater levels of telomere ssDNA from TARG1-deficient cells (Fig. 1a). Notably, ADP-ribose signals were only detected in telomeric DNA captured from TARG1-knockout (KO) cells. The telomere DNA-ADPr signals were DNaseI sensitive (Fig. 1a) and resistant to RNaseA. DNA-ADPr was also detected in samples isolated by telomere RSE from TARG1-deficient IMR90 human fibroblasts that were immortalized with HPV E6/E7 oncoproteins (IMR90^E6/E7), as well as in TARG1-deficient HeLa cells (Fig. 1a). U2OS and HeLa cells activate alternative lengthening of telomeres or telomerase-mediated telomere extension mechanisms, respectively. IMR90^E6/E7 cells lack a telomere extension mechanism. Therefore, DNA-ADPr may be a general feature of telomeres.

Reconstituting U2OS TARG1-KO cells with green fluorescent protein (GFP)-tagged WT-TARG1 removed the ADPr signals from telomere DNA. In contrast, the DNA-ADPr signals were not altered on expression of the TARG1 ADP-ribose hydrolysis defective mutant (TARG1-K84A)[12,20] (Fig. 1b). By in vitro hydrolase assay, we found that purified full-length TARG1 protein completely removed ADP-ribose from telomere DNA isolated from TARG1-KO cells, while the TARG1-K84A mutant protein did not. This contrasted with the partial in vitro hydrolysis of ADP-ribose either human PARG or bacterial DarG (Extended Data Fig. 1b). This was despite cellular observations in which PARG inhibition (PARGi) stimulated telomere DNA-ADPr in control U2OS cells and enhanced the DNA-ADPr signals detected in TARG1-KO cells, as well as TARG1-WT/K84A complemented cells (Fig. 1b). This can be explained by biochemical studies

demonstrating that PARG specifically removes protein-linked PAR chains but that the terminal mono-ADP-ribose (MAR) moiety is removed by specialized MAR hydrolases[21]. Since TARG1 is capable of cleaving MAR, the additive effect of PARGi on the levels of telomeric DNA-ADPr detected in TARG1-KO cells likely reflects the removal of PAR from DNA by PARG to limit the excessive accumulation of toxic PAR chains[21].

We next determined which PARP(s) mediate telomere DNA-ADPr. Olaparib (PARPi) completely abolished the signals in TARG1-KO and TARG1-K84A expressing U2OS cells implicating PARP1 and PARP2 (Fig. 1b). While PARP2 depletion partially reduced telomere DNA-ADPr, PARP1 knockdown abolished virtually all the ADPr signal from telomere DNA samples (Extended Data Fig. 1c). Depletion of HPF1 (histone PARylation factor 1), a cofactor that directs PARP1-2 dependent serine-ADPr of histone H3 and chromatin during the DNA damage response (DDR)[22] and ARH3 (ADP-ribosyl-hydrolase 3) a serine-ADPr hydrolase that counteracts PARP1/2-HPF1 (ref. 23) did not affect telomere DNA-ADPr, effectively ruling out their possible contribution (Extended Data Fig. 1c). As with Olaparib treatment and PARP1 knockdown, telomere DNA-ADPr was also not detected in TARG1-PARP1 deficient U2OS cells (Fig. 1c). Introducing WT-PARP1 restored telomere DNA-ADPr. By contrast, the catalytic-dead PARP1-EQHA2 (E988Q, H862A) mutant, did not[24]. However, expressing PARP1-EQ, a PARP1 mutant that catalyzes MAR but is incapable of PAR chain extension[25], restored the DNA-ADPr signal intensity nearly to wild-type (WT) levels as indicated by using recently described MAR-specific antibodies[26] (Fig. 1c and Extended Data Fig. 1d). From these combined in vitro and cellular experiments, we conclude that PARP1 is the major catalyst of telomeric DNA-ADPr and that TARG1 is the primary hydrolase responsible for removing ADP-ribose from telomere DNA.

### DNA break and S-phase accumulation of telomeric DNA-ADPr

We next asked under which physiological conditions telomere DNA-ADPr is stimulated. First, we exposed control and TARG1-KO U2OS cells to the global genotoxic damaging agent hydrogen peroxide and used TRF1-FokI, Cas9 D10A and FAP-TRF1 to introduce targeted double-strand (ds) DNA breaks, single-strand (ss) DNA breaks and singlet oxygen ($^1O_2$) production specifically within telomeres (Extended Data Fig. 1e)[27,28]. Except for FAP-TRF1, which generates 8-oxo guanine base lesions at telomeres, global and localized telomere DNA damage induced high levels of protein-ADPr and DDR signaling (Extended Data Fig. 1f), but also enhanced telomere DNA-ADPr in TARG1-KO cells (Fig. 1d). This implied that ss- and dsDNA breaks can acutely stimulate telomeric DNA-ADPr.

Previous studies suggested endogenous (that is, DNA damage independent) protein-ADPr oscillates across the cell cycle, peaking in S-phase[21,29,30]. To examine whether the same patterns are associated with telomere DNA-ADPr, U2OS cells were synchronized in G0 and G1-S by serum starvation or double-thymidine block, respectively, and released G1/S arrested cells into mid-S-phase (Fig. 1e). Efficient cell cycle synchronization was verified by monitoring Cyclin E expression (Fig. 1e) and flow cytometry (Extended Data Fig. 1g). Using the RSE assay, we found that telomeric DNA-ADPr accumulates in S-phase (Fig. 1e). Furthermore, we found that stalling S-phase progression by hydroxyurea (HU)-mediated nucleotide deprivation and intra-S checkpoint activation (Extended Data Fig. 1h) abolished telomeric DNA-ADPr (Fig. 1f). In contrast, acute (1 h) ATR inhibition (ATRi) that provokes replication origin firing[31] increased telomeric DNA-ADPr (Fig. 1f). These results provided strong evidence that telomeric DNA-ADPr is associated with DNA replication in S-phase.

### DNA-ADPr during telomeric lagging strand maturation

To further define the nature of telomeric DNA-ADPr in S-phase, we applied cesium chloride (CsCl) density gradient centrifugation of IdU (5′-Iodo-2-deoxyuridine)-pulsed U2OS cells treated with PARGi to differentially separate nascent leading and lagging telomeric DNA

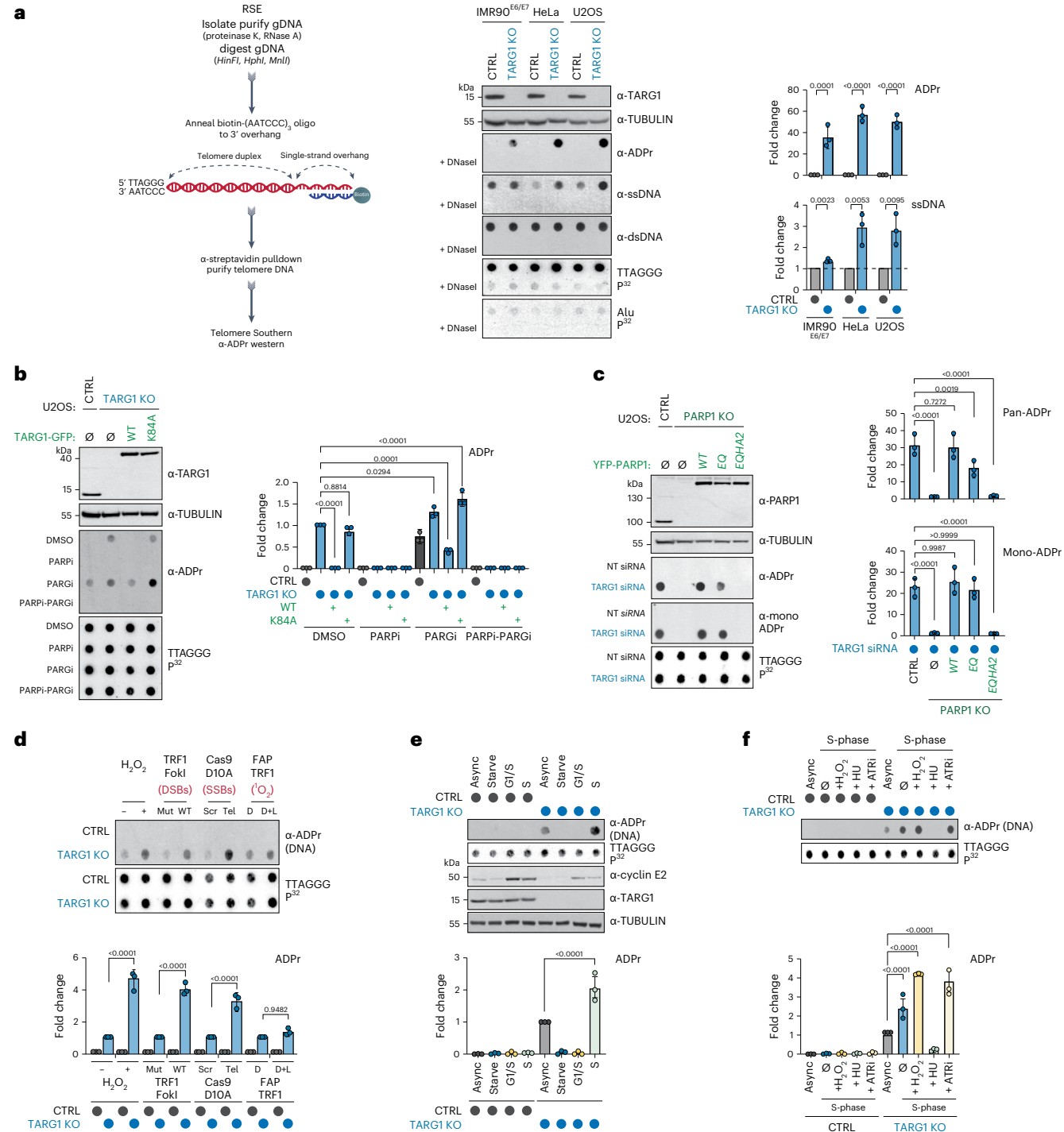

**Fig. 1 | TARG1 mediated ADP-ribosylation of telomere DNA. a**, RSE methodology (left), western blot, telomeric western dot blot of ADPr, ssDNA, dsDNA and Southern blot of telomeric DNA and Alu repeats of CTRL and TARG1-KO IMR90$^{E6/E7}$, HeLa and U2OS cells (middle) and quantification of ADPr and ssDNA normalized to the CTRL line (right). **b**, Western blot and telomeric dot blot of ADPr and Southern blot of telomeric DNA (left) of U2OS CTRL, TARG1-KO and TARG1-KO cells with expression of either GFP-tagged full-length (WT) TARG1 or TARG1-K84A treated with dimethylsulfoxide (DMSO), PARPi, PARGi or PARPi and PARGi and quantification of ADPr of the samples normalized to TARG1-KO + DMSO (right). **c**, Western blot and telomeric dot blot of ADPr and mono-ADPr and Southern blot of telomeric DNA (left) of U2OS CTRL, PARP1-KO and PARP1-KO cells with inducible expression of YFP-tagged full-length (WT) PARP1, PARP1 E998Q or PARP1-EQHA2 after control or TARG1 knockdown and quantification of pan-ADPr and mono-ADPr (right). Quantifications are normalized to U2OS CTRL siCTRL (data not shown). **d**, Telomeric western dot

blot and Southern blot of telomeric DNA (top) and ADPr quantification (bottom) of U2OS CTRL and TARG1-KO treated with $H_2O_2$, transiently transfected with TRF1-FokI (D450A or WT), Cas9 D10A (scr or tel) or FAP-TRF1 cells treated with dye or dye and light. Each quantification is normalized to control conditions. **e**, Telomeric western dot blot of ADPr and Southern blot of telomeric DNA and western blot (top) and ADPr quantification (bottom) of U2OS CTRL and TARG1-KO in asynchronous (Async), serum-starved (starve), G1/S or S phases. Quantification is normalized to TARG1-KO (async). **f**, Telomeric DNA-ADPr dot blot (top) and ADPr quantification (bottom) of U2OS CTRL and TARG1-KO in either asynchronous (async) or S-phase treated with $H_2O_2$ (2 mM, 15 min), hydroxyurea (HU) (2 mM, 1 h) or ATR inhibitor (ATRi) (5 nM, 1 h). Quantification is normalized to TARG1-KO (async). Mean and s.e.m. are shown from three independent experiments in **a**–**f**, and groups were compared with a one-way ANOVA followed by Tukey's multiple-comparisons test for pairwise comparisons.

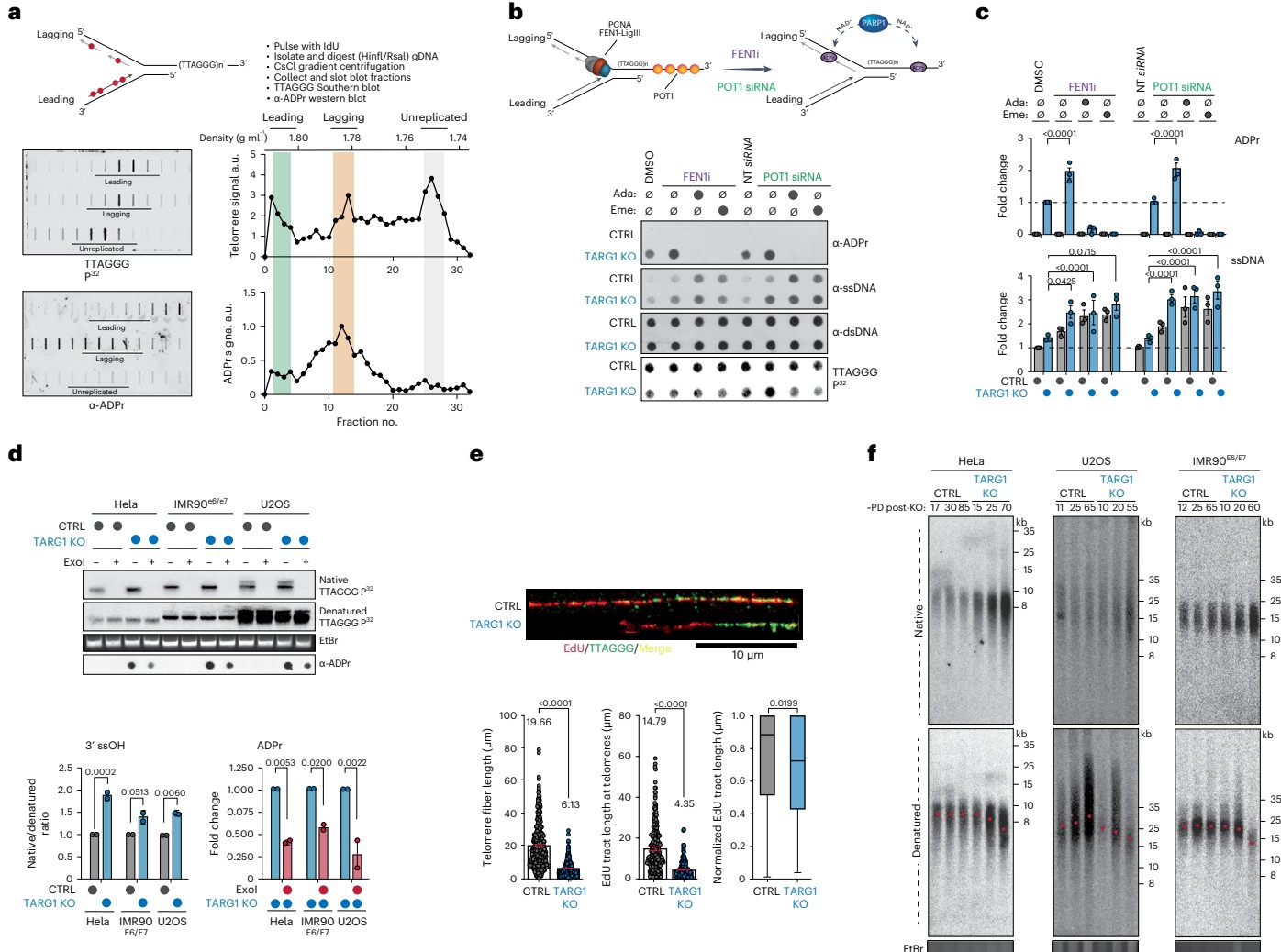

**Fig. 2 | DNA-ADPr is associated with lagging strand telomere replication.**
**a**, CsCl density gradient centrifugation of IdU-pulsed U2OS TARG1-KO cells methodology (top) to separate leading and lagging telomeric DNA strands. Southern blot of telomeric DNA and telomeric western ADPr slot blot (bottom left) with quantification of the corresponding strands (bottom right). gDNA, genomic DNA. a.u., arbitrary units. **b**, Schematic of effects of POT1 knockdown (siPOT1) or FEN1 inhibition (FEN1i) (top). Telomeric western dot blot of ADPr, ssDNA, dsDNA and Southern blot of telomeric DNA of U2OS CTRL and TARG1-KO with either FEN1i or siPOT1 treated with adarotene (Ada) or emetine (Eme). **c**, Quantification of ADPr (top) or ssDNA (bottom). Quantification of ADPr and ssDNA dot blots (right) normalized to TARG1-KO + DMSO and CTRL + DMSO, respectively. Data represent the mean ± s.e.m., $n = 3$ biological replicates and groups were compared with a one-way ANOVA followed by Tukey's multiple-comparisons test for pairwise comparisons. **d**, DNA from CTRL and TARG1-KO HeLa, IMR90$^{E6/E7}$ and U2OS cells treated with exoI, ran on a gel and probed for telomeric DNA in native and denaturing Southern blots. Telomeric western dot blot for ADPr (bottom). Quantification of the native/denatured telomeric DNA ratio (left) normalized to corresponding HeLa, IMR90$^{E6/E7}$ or U2OS CTRL line and quantification of ADPr dot blot (right) normalized to TARG1-KO without exoI treatment for each corresponding cell line. Data represent the mean ± s.e.m., $n = 2$ biological replicates and groups were compared with a one-way ANOVA followed by Tukey's multiple-comparisons test for pairwise comparisons. **e**, SMAT-representative fibers (top) and quantification of telomere fiber length (left), EdU tract length at telomeres (middle) and normalized EdU tract length (right). Data represent the mean ± s.e.m. of three biological replicates (>400 fibers scored (two-tailed Mann–Whitney test)) (left and middle). Data on the right are shown in box and whisker format with minimum and maximum values and median line. **f**, Telomere length analysis by PFGE of CTRL and TARG1-KO HeLa, U2OS and IMR90$^{E6/E7}$ cells with indicated population doublings (PDs) following transfection with Cas9 and single-guide RNAs. Red dots indicate mean telomere lengths.

strands (Fig. 2a). We chose to use acute PARG inhibition as it promotes DNA-ADPr without adversely interfering with cell cycle dynamics (Fig. 1c)[32]. Fractionated DNAs corresponding to leading, lagging and unreplicated telomeres were slot blotted and probed with telomere probes in Southern blot and western blot using ADPr antibodies. This uncovered that telomere DNA-ADPr occurs selectively on the lagging telomere DNA strand (Fig. 2a).

PARP1-mediated ADP-ribosylation has been linked with controlling the maturation of nascent DNA strands and the rate of DNA replication[33,34]. In imaging experiments, it was shown that FEN1

(Flap Endonuclease 1) inhibition and depletion, which prevents cleavage of the 5′ DNA-RNA flap from nascent Okazaki fragments formed during lagging strand synthesis, increased S-phase ADPr and exacerbated PARP inhibitor cytotoxicity[35]. These studies provided strong evidence for unligated Okazaki fragments as the primary sources of S-phase ADPr. In agreement with this, we found that FEN1 inhibition (FEN1i) enhanced telomeric DNA-ADPr in TARG1-KO U2OS cells (Fig. 2b,c). Knockdown of FEN1 and DNA ligase I (LigI), which ligates Okazaki fragments, also increased elevated telomeric DNA-ADPr signals (Extended Data Fig. 2a–c). These increases in telomere DNA-ADPr

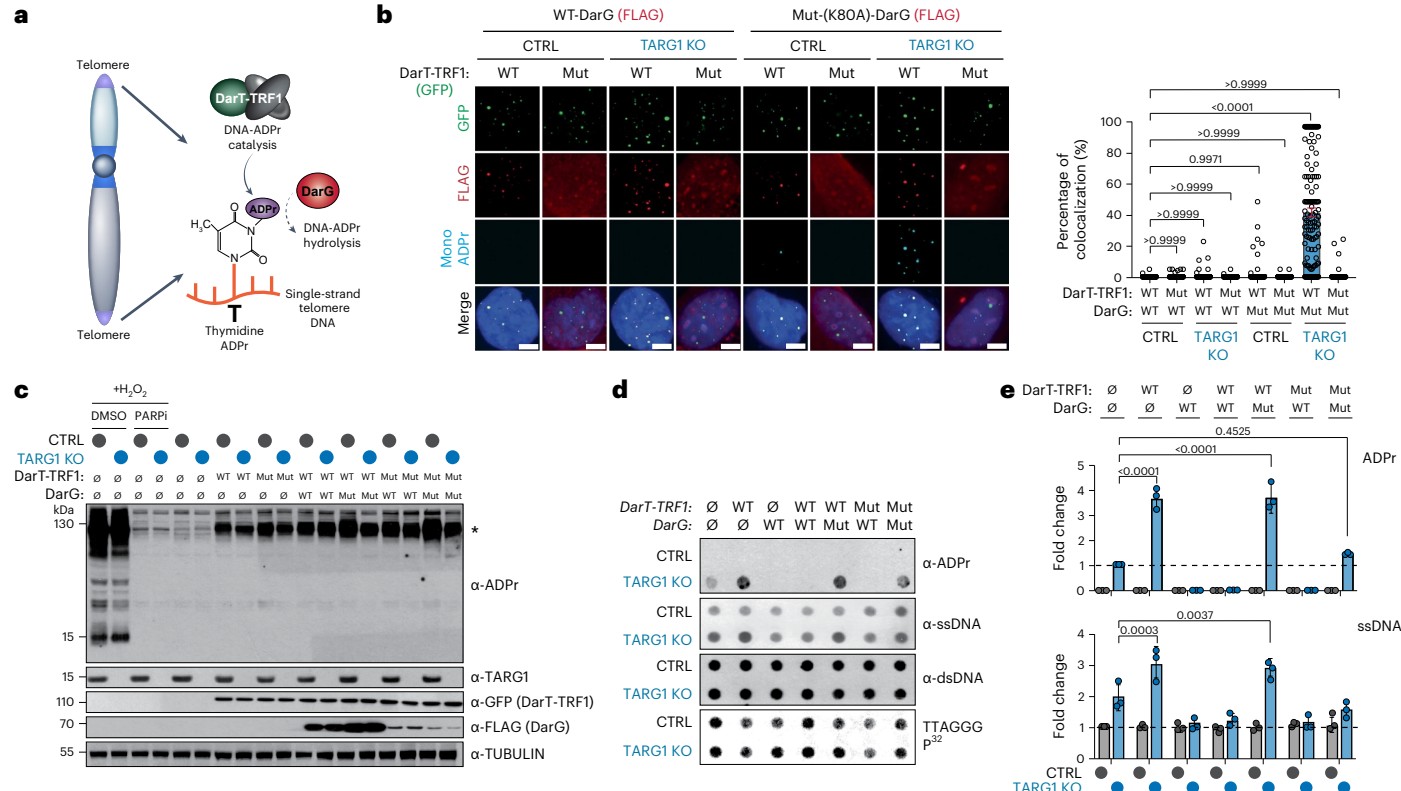

**Fig. 3 | Direct targeting of telomeres via a DarT-TRF1 endotoxin fusion.**
**a**, Schematic of GFP-tagged DarT-TRF1-induced telomeric DNA ADP-ribosylation and subsequent removal by DarG (red). **b**, Immunofluorescence of U2OS CTRL and TARG1-KO cells after expression of GFP-tagged WT-DarT-TRF1 (WT) or E160A DarT-TRF1 (mut) and FLAG-tagged WT-DarG or K80A DarG (left) and quantification of colocalization of mono-ADPr and GFP-positive telomeres (right). In all conditions, more than 140 cells were analyzed. Scale bars represent 10 μm. **c**, Western blot of U2OS CTRL and TARG1-KO cells after expression of GFP-tagged WT-DarT-TRF1 (WT) or E160A DarT-TRF1 (mut) and FLAG-tagged WT-DarG

(WT) or K80A DarG (mut). U2OS CTRL and TARG1-KO were treated with 2 mM $H_2O_2$ and 2 mM $H_2O_2$ with PARPi for positive and negative controls, respectively. **d**, Telomeric western dot blot of ADPr, ssDNA, dsDNA and Southern blot of telomeric DNA. **e**, Quantification of ADPr (top) normalized to TARG1-KO and ssDNA (bottom) normalized to TARG KO and CTRL, respectively. after expression of vectors in **c**. Mean and s.e.m. are shown from three independent experiments in **b** and **e**, and groups were compared with a one-way ANOVA followed by Tukey's multiple-comparisons test for pairwise comparisons.

correlated with ss telomere DNA abundance in TARG1-KO cells (Fig. 2b,c and Extended Data Fig. 2b). The FEN1i-induced DNA-ADPr signals were completely ablated after incubating TARG1-KO cells with adarotene or emetine, potent inhibitors of the POLA1 catalytic subunit of DNA Polα[36] and lagging strand synthesis[37], respectively (Fig. 2b,c). These results indicate that DNA-ADPr is coupled to DNA synthesis on the lagging telomere strand, potentially accumulating at ssDNA discontinuities between unligated Okazaki fragments.

Whereas FEN1 and DNA LigI regulate global lagging strand synthesis, POT1 is a Shelterin complex subunit that binds the single-stranded TTAGGG-rich 3′ overhang and coordinates telomere replication. As with interfering with FEN1-mediated Okazaki fragment maturation, depleting POT1 caused elevated DNA-ADPr that again was sensitive to adarotene and emetine (Fig. 2b,c and Extended Data Fig. 2a). This reinforced the premise that DNA-ADPr accumulation is associated with lagging strand synthesis. However, POT1 also binds the ssDNA 3′ overhang and recruits the CST (CTC1-STN1-TEN1) complex that mediates fill-in DNA synthesis by Polα-Primase[38,39]. POT1 disruption causes increased telomeric ssDNA 3′ overhangs. This potentially accounted for the increased ssDNA detected at telomeres following POT1 depletion in TARG1-KO cells (Fig. 2b,c). We conducted in-gel Southern blot using radiolabeled telomere probes, this time under native conditions to detect the single-stranded 3′ telomere overhang in HeLa, IMR90[E6/E7] and U2OS control and TARG1-KO cells (Fig. 2d). The sensitivity of the signals to bacterial 3′ to 5′ exonuclease (exoI) digestion confirmed

that they represent the 3′ overhangs (Fig. 2d). We then denatured and reprobed the same gel to calculate the amount of ssDNA overhang relative to the total telomeric DNA content. This revealed a roughly twofold increase in the ratio of ssDNA corresponding to the 3′ overhang relative to duplex telomeric DNA in the TARG1-deficient cell lines examined (Fig. 2d). Furthermore, expressing WT-TARG1 restored the normal 1/1 ratio of overhang to duplex telomere DNA in U2OS (Extended Data Fig. 2d). The effect of TARG1 on the abundance of ssDNA telomeric DNA corresponding to the overhang raised the question of whether the overhang itself could be ADP ribosylated. We found that parallel exoI treatment of telomere RSE samples from these TARG1-KO cell lines significantly reduced DNA-ADPr (Fig. 2d) indicating that the 3′ ssDNA overhang accounts for a substantial (50–70%) portion of the total ADPr signals detected.

Using conditional POT1-KO 293E cells that can be complemented with either WT or mutant POT1 (Extended Data Fig. 2e,f), we again observed increased telomere DNA-ADPr and ssDNA after depletion of TARG1 (Extended Data Fig. 2g,h). This increase was suppressed following complementation with WT POT1. Similarly, expression of POT1-R83E, a newly described POT1 mutant that cannot protect the 5′ recessed end at the ds–ss-DNA junction from recognition by the DNA damage machinery[40] strongly suppressed the accumulation of telomere DNA-ADPr in TARG1-deficient POT1-KO cells. However, the expression of POT1-F62A, a POT1 ssDNA binding mutant whose expression hyperextends the overhang[41] (Extended Data Fig. 2g,h)

did not suppress DNA-ADPr as efficiently (Extended Data Fig. 2e,f). Collectively, these experiments clarified further that exposed ssDNA is the preferred substrate for PARP1-mediated DNA-ADPr, whose timely degradation is mediated by TARG1.

## Impaired telomere replication due to TARG1 deficiency

The persistence of unhydrolyzed DNA-ADPr due to the absence of TARG1 could represent an obstacle that compromises telomere replication. By conducting SMAT (single-molecule analysis of telomeres), a DNA fiber assay adapted for telomeres, we first observed that the mean telomere fiber length was considerably shorter in TARG1-KO U2OS cells (Fig. 2e). In measuring the ratio of telomere fiber length to the length of fibers labeled with EdU (5-ethynyl-2-deoxyuridine), we determined that net telomere DNA synthesis is considerably reduced in TARG1-KO cells indicative of impaired telomere replication in those cells (Fig. 2e). Following up on the apparent significantly shorter telomere DNA fibers in TARG1-KO U2OS cells, we conducted classical telomere length analysis by pulsed-field gel electrophoresis (PFGE) and Southern blot of DNA from control and TARG1-KO HeLa, U2OS and IMR90$^{E6/E7}$ cells that were cultured over successive weeks. Under native conditions, we observed progressive, robust increases in the signal intensity in DNA from the TARG1-KO cell lines (Fig. 2f). This confirmed previous results showing increased ssDNA in the RSE samples from TARG1-KO cells. Following denaturation of the same gel, robust telomere shortening was evident in TARG1-KO HeLa, U2OS and IMR90$^{E6/E7}$ cells (Fig. 2f). The extent of telomere shortening varied from moderate in TARG1-KO IMR90$^{E6/E7}$ and HeLa to extensive in TARG1-KO U2OS cells. These variations may be due to cell line intrinsic telomere shortening rates following incomplete replication of the lagging strand and potential interference in overhang management that impairs fill-in DNA synthesis by the CST complex[39]. We did not detect differential 53BP1 accumulation at telomeres in asynchronous control or TARG1-KO U2OS cells. There was, however, a twofold increase in 53BP1 positive telomeres in S-phase synchronized TARG1-KO cells. The modesty of this effect, however, indicates that unhydrolyzed DNA-ADPr does not elicit a robust DDR (Extended Data Fig. 2i). This may be linked to the preferential accumulation of DNA-ADPr on ssDNA (Extended Data Fig. 2g,h), which is a less potent activator of ATR DNA damage-induced signaling than exposed ends[42]. However, more micronuclei, particularly those containing telomeric DNA fragments and Replication Protein A (RPA), which binds to ssDNA, were observed in TARG1-KO U2OS cells (Extended Data Fig. 2j). The accumulation of these by-products that often form during mitosis demonstrates the enhanced telomere and chromosomal instability due to TARG1 deficiency. These data indicate that removing DNA-linked ADPr from ssDNA, including at the 3′ overhang, is required for efficient telomere replication and telomere length management, thereby contributing to genome integrity.

## Direct targeting of DNA-ADPr to telomeres

Considering the pervasive contribution of PARP1 in DNA damage repair and replication, we could not be certain that these defects and alterations in telomere replication and length were truly due to PARP1-directed DNA-ADPr. Yet, discriminating PARP1's protein and DNA modifying activities is a major obstacle to advancing our understanding of the physiological impact of deregulated DNA-ADPr. No separation of function PARP1 mutant has been identified and will prove challenging. However, we devised an experimental system that enables the direct ADPr of telomeric DNA, without stimulating PARP1 or its protein-ADPr activity.

*Thermus aquaticus* DarT (also known as DarT2) is a PARP1-like enzyme that catalyzes mono-ADPr of thymidine bases in ssDNA[9,12,43]. DarT shows no activity on dsDNA and is incapable of protein or RNA-ADPr[13]. We targeted GFP-tagged DarT to telomeres by fusing it with the telomere-sequence binding protein, TRF1 (Fig. 3a). We also generated a mutant DarT-TRF1 fusion lacking its ADP-ribosyl-transferase activity, DarT-TRF1-E160A. By immunofluorescence, both WT and mutant DarT-TRF1 localized to roughly 80% of detectable telomeres marked by TRF2 (Extended Data Fig. 3a). Mono-ADPr foci were at telomeres only in TARG1-KO U2OS cells expressing WT-DarT-TRF1 (Extended Data Fig. 3b). Both DarT-TRF1 localization and mono-ADPr foci were unaltered by PARPi (Extended Data Fig. 3b). We also used FLAG-tagged *T. aquaticus* DarG, which hydrolyzes DarT-generated thymidine-linked DNA-ADPr, as well as a catalytically inactive mutant, DarG-K80A[13]. WT-DarG completely suppressed MAR foci formation by DarT-TRF1 in TARG1-KO cells. In contrast, telomeric MAR foci were readily detected following expression of catalytically inactive-DarG (Fig. 3b). We observed reduced expression of DarG when mutant DarT was co-expressed in cells (Fig. 3c). This is consistent with structural studies revealing DarT–DarG interactions that can be disrupted by mutation of the catalytic domain of DarT[43]. Western blot analysis confirmed that in contrast to $H_2O_2$ treatment, characteristic protein-ADPr smears were not observed following the expression of DarT-TRF1 in control or TARG1-KO U2OS cells, indicating that it does not trigger widespread protein or histone ADPr (Fig. 3c).

By RSE and ADPr dot blot assay, we observed that WT-DarT-TRF1 stimulated telomere DNA-ADPr above the baseline DNA-ADPr catalyzed by PARP1 in TARG1-KO U2OS cells (Fig. 3d,e). Notably, the expression of WT-DarG or WT-TARG1 suppressed all telomere DNA-ADPr in TARG1-KO U2OS cells (Fig. 3d,e and Extended Data Fig. 3c). This result is particularly significant since DarG exhibits selectivity for thymidine-linked ADPr hydrolase activity on ssDNA substrates[8,9,12]. Therefore, base-linked ADPr on ssDNA is likely to be the major species of DNA-ADPr present at telomeres that TARG1 removes. Last, the stimulation of ADPr by DarT-TRF1 again coincided with increased telomere ssDNA (Fig. 3d,e). We interpreted this to imply that DarT-TRF1 mediated DNA-ADPr on the lagging strand could stall or uncouple replication of the telomere DNA strands, thereby increasing the availability of ssDNA that could also provide an additional substrate for DarT-TRF1 dependent DNA-ADPr. In agreement, we observed that FEN1i and POT1 depletion enhanced telomere ADPr by WT-DarT-TRF1, but this was suppressed by both adarotene and emetine (Extended Data Fig. 3d). These experiments demonstrate that the DarT-TRF1 system can catalyze DNA-ADPr of thymidine bases in telomeric ssDNA during telomere replication.

**Fig. 4 | Unhydrolyzed telomere DNA-ADPr impairs telomere integrity.**
**a**, Representative SMAT fibers (left) of U2OS CTRL and TARG1-KO cells after expression of WT-DarT-TRF1 (WT) or E160A DarT-TRF1 (mut), quantification of EdU tract length at telomeres (middle) and normalized EdU tract length (right). Mean and s.e.m. (middle) and a box and whisker plot with minimum and maximum values and median line (right). Results shown are from three independent experiments and groups were compared with Mann–Whitney tests (>200 fibers scored per condition). **b**, Immunofluorescence images of U2OS CTRL and TARG1-KO cells (left) after expression of GFP-tagged WT-DarT-TRF1 (WT) or E160A DarT-TRF1 (mut) and quantification of percentage of RPA2 + GFP-positive telomeres (right) with more than 150 cells analyzed per condition. Scale bars represent 10 μm. **c**, CO-FISH representative images of U2OS CTRL and TARG1-KO chromosomes after dox-inducible expression of either GFP-tagged WT-DarT-TRF1 (WT) or E160A DarT-TRF1 (mut). **d**, CO-FISH quantification of percentage leading (red) and lagging (green) fragile telomeres (left) and percentage telomere sister chromatid exchanges (telomere SCEs) (right) of images in **c** (>5,000 telomeres scored per condition). **e**, Colony formation assay representative images of U2OS CTRL and TARG1-KO cells after transient expression of either GFP-tagged WT-DarT-TRF1 (WT) or E160A DarT-TRF1 (mut) and FLAG-tagged WT-DarG (WT) or K80A DarG (mut). **f**, Quantification of the number of colonies from **e** normalized to CTRL. Mean and s.e.m. are shown from three independent experiments in **b**, **d** and **f**, and groups were compared with a one-way ANOVA followed by Tukey's multiple-comparisons test for pairwise comparisons.

## Unhydrolyzed DNA-ADPr compromises telomere integrity

Phenotypically, we first sought to assess whether telomere-specific DarT-TRF1 induced DNA-ADPr influenced telomere replication by again performing the SMAT assay. Here, we detected moderately shorter tracts of EdU-labeled telomere DNA fibers following the expression of WT-DarT-TRF1 compared with the catalytically inactive DarT-TRF1-E160A

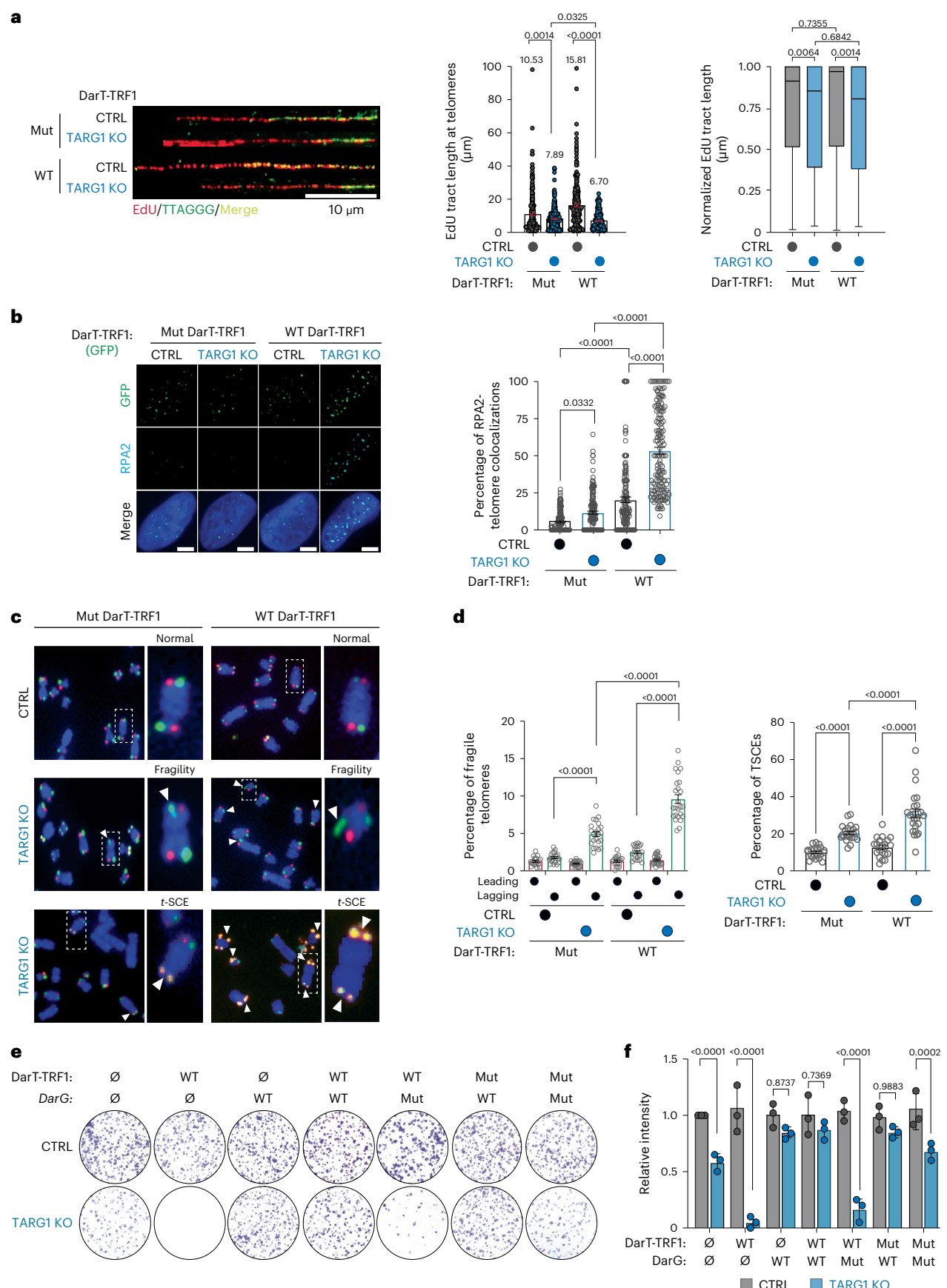

mutant in TARG1-KO U2OS cells (Fig. 4a). Normalization of the EdU tract lengths to telomere fiber lengths revealed that WT-DarT-TRF1 expression elicited a modest net reduction in telomere DNA synthesis (Fig. 4a). Yet, we independently corroborated the negative effect of WT-DarT-TRF1 on telomere replication by monitoring EdU incorporation directly at telomeres in U2OS cells (Extended Data Fig. 4a).

The perturbations in telomere replication caused by DarT-TRF1-induced ADPr could lead to the exposure of ssDNA that would be recognized and bound by the RPA complex. In agreement, we observed more RPA2 foci colocalizing with telomeres by immunofluorescence in TARG1-KO cells expressing WT-DarT-TRF1 (Fig. 4b). This was despite acute WT-DarT-TRF1 expression stimulating RPA2 and H2AX phosphorylation, but not phosphorylation of CHK1-serine 317 by ATR kinase (Extended Data Fig. 4b) or significantly altering cell cycle phasing (Extended Data Fig. 4c). Furthermore, we expected that the unhydrolyzed ADPr clusters and defects in lagging strand synthesis could lead to the fragile telomere phenotype[44]. Telomere fragility is associated with replicative complications at telomeres and can be visualized as discontinuous telomere fluorescence in situ hybridization (FISH) signals on sister chromatids of metaphase chromosomes[45]. We conducted chromosome-orientation-FISH (CO-FISH) experiments to differentially label leading and lagging telomere strands on metaphase chromosomes and observed elevated frequency of telomere fragility that was more pronounced on lagging strand telomeres in TARG1-KO U2OS and IMR90[E6/E7] cells (Extended Data Fig. 4d,e). Furthermore, we observed that this fragile telomere phenotype was significantly exacerbated following the expression of WT-DarT-TRF1 (Fig. 4c,d). In addition to telomere fragility, we found that the TARG1-deficient cell lines exhibited a clear increase in telomere sister chromatid exchanges (Extended Data Fig. 4d,e), which are generated through replicative stress-associated recombination to salvage stalled replication forks. As with telomere fragility, the frequency of telomere sister chromatid exchanges was enhanced following the expression of WT-DarT-TRF1, but not its mutant counterpart (Fig. 4c,d).

These results highlight the adverse affects of acute DarT1-TRF1 expression and unhydrolyzed DNA-ADPr on telomere integrity, and prompted us to address how the constitutive induction of DarT-TRF1-mediated telomere DNA-ADPr might affect cell viability. In colony formation assays, we found that the parental TARG1-KO cells exhibited moderately impaired colony formation compared to their control counterparts (Fig. 4e,f). However, WT-DarT-TRF1 expression completely ablated colony formation of TARG1-KO cells (Fig. 4e,f). The cytotoxicity of TARG1-KO cells was rescued by co-expression of DarG and DarT-TRF1. In contrast, mutant DarG did not influence colony formation (Fig. 4e,f). By using the DarT-TRF1 experimental tool, we have been able to ascertain the specific molecular effects of DNA-ADPr linked to thymidine bases, which, left unhydrolyzed due to TARG1 deficiency, diminishes the integrity of telomeres due to impaired replication and ultimately affects cell survival.

## Discussion

Here we identified that telomeric DNA repeat sequences are subject to PARP1-mediated ADP-ribosylation that is reversed by the ADPr hydrolase, TARG1. That TARG1 must be disrupted to detect DNA-ADPr reflects that, like the protein poly-ADP-ribosylation, telomere DNA-ADPr is highly dynamic and subject to stringent control by TARG1. Notably, telomere DNA-ADPr is enriched during S-phase and appears to be coupled to DNA replication. Previous studies have implicated PARP1 and the S-phase accumulation of ADPr species at DNA replication sites with controlling Okazaki fragment maturation and ligation[33,35]. Other studies showed that PARP1 inhibition drastically deregulates replication fork dynamics, causing fork acceleration, replication stress and genome instability[34]. Along similar lines, our study revealed that telomere DNA-ADPr is functionally and intricately linked with lagging strand telomere replication, and the failure to remove this modification

due to TARG1 deficiency impairs telomere replication. Consistent with those previous studies, we found that interfering with FEN1-mediated ligation of Okazaki fragments or telomere replication by depleting POT1 robustly stimulated telomere DNA-ADPr. In these instances, the presence and stimulation of telomere DNA-ADPr coincided or was accompanied by greater levels of telomeric ssDNA. DNA-ADPr at unligated Okazaki fragments could uncouple the replication of telomere DNA strands, leading to ssDNA accumulation at stalled replication forks in telomeres. The availability of more ssDNA could recruit PARP1 and thus concentrate more DNA-ADPr at telomeric replication forks. Thus, the prolonged presence of unhydrolyzed DNA-ADPr at sites of misprocessed Okazaki fragments could cause postreplicative gaps in telomeres that would explain the elevated telomere fragility phenotype in TARG1-KO cells (Extended Data Fig. 5).

Most notably, our study revealed that the single-stranded 3′ overhang, a defining structural feature of telomeres, is also a substrate for DNA-ADPr. By disrupting POT1-mediated regulation of the telomere overhang, we again found that the increased availability of ssDNA and not merely exposing the 5′ recessed DNA end stimulates DNA-ADPr. The presence of ADPr could interfere with the placement of the terminal Okazaki fragments thereby increasing the G-rich overhang. Such incomplete lagging strand synthesis could cause the telomere shortening that we observed in the TARG1-KO cell lines (Extended Data Fig. 5). Alternatively, it is well established that the lengthening of the overhang following disrupting POT1 binding to the overhang is in large part due to impaired recruitment of CST-Polα that mediates C-strand fill-in[38,39]. The increased overhang and ADPr signal detected in TARG1-KO cells could reflect impaired C-strand fill-in in these cells due to persistent DNA-ADPr (Extended Data Fig. 5). While both scenarios discussed above could occur, implicit in each is that the telomere binding of POT1 and/or other factors that regulate telomere replication might be prevented, even transiently, due to DNA-ADPr. Whether POT1 or CST recruitment to telomeres is impaired by DNA-ADPr in TARG1-KO cells is unclear (Extended Data Fig. 5). But, if so, it could implicate DNA-ADPr as affecting protein dynamics at telomeres. Adding negatively charged ADPr on ssDNA might interfere with stepwise protein localization, as shown for protein-ADPr[46]. But whether the chemical and biophysical properties of DNA-ADPr influence DNA binding properties or protein localization and recruitment in the same manner as protein-ADPr is a key unresolved question.

The natural availability of ssDNA and structured ssDNA substrates (G4s, R-loops) for PARP1's DNA modifying activity at telomeres might explain why replicative defects stemming from unhydrolyzed ADPr due to TARG1 deficiency are more pronounced at telomeres. Furthermore, our ability to manipulate DNA-ADPr catalysis specifically at telomeres with DarT or DarG enabled us to uncover a direct role for the regulated catalysis and degradation of modification in maintaining telomere replication, but also to reveal the consequences of DNA-ADPr dysregulation on telomere, and thereby genome integrity. However, DNA-ADPr is likely more frequently catalyzed genome-wide by PARP1 due to erroneous or abortive Okazaki fragment extension by Polδ dependent strand displacement synthesis or defective RNA primer and 5′ flap removal. Like at telomeres, the timely removal of DNA-ADPr by TARG1 could be crucial for global DNA replication fidelity[33,35]. Persistent replicative stress and pathological ssDNA due to TARG1 inhibition could benefit strategies that interfere with cellular ADPr (that is, PARP inhibitors) to eliminate cancerous cells[35,36,47]. Furthermore, our observations that DNA-ADPr accumulates at DNA strand breaks and nicks as well as telomeres in TARG1-deficient cells implicate telomere and nontelomeric DNA-ADPr in human disease and its treatment with ADPr-modulating therapeutics. We expect that the fundamental lessons of DNA-ADPr acquired through this investigation of telomeres will have broad implications for our understanding of the cellular function of DNA-ADPr and the contribution of ADPr regulation in genome stability.

## Online content

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

## Methods

### Cell culture

U2OS, HeLa and IMR90-E6/E7 cell lines were obtained, authenticated by short tandem repeat profiling through the American Type Culture Collection and routinely tested for mycoplasma with MycoAlert (Lonza). Doxycycline-inducible PARP1 and PARP1-KO lines were generated in I. Ahel's laboratory. U2OS and HeLa cell lines were cultured in DMEM supplemented with 10% bovine growth serum. Cells were cultured at atmospheric oxygen conditions of 20% $O_2$ and 7.5% $CO_2$ at 37 °C. The IMR90-E6/E7 cell line was cultured with 10% DMEM supplemented with 10% fetal calf serum in 5% $O_2$ and 7.5% $CO_2$ at 37 °C. HEK293E POT1-KO and POT1 complemented cells were obtained from and validated by J. Nandakumar's laboratory.

### Generation of TARG1-KO cell lines

U2OS, HeLa and IMR90-E6/E7, control (CTRL) and TARG1-KO, cell lines were engineered using the Synthego CRISPR editing kit with RNPs (UUUUAGAUCACUUAUGUGAA, ACAAUCCUCACUGAUACAGU, GACAAUCCUCACUGAUACAG and UUUAGCCCACUGUAUCAGUG). U2OS TARG1 rescue cell lines were created by transfecting pDEST47 TARG1 and pDEST47 TARG1-K84A into U2OS TARG1-KO cells. The disruption of TARG1 protein expression was confirmed by western blot.

### Generation of FAP-TRF1 TARG1-KO cell line

U2OS CTRL and TARG1-KO cells were transduced with pLVX-FAP-mCer-TRF1 (ref. [16]) and selected with 500 µg ml⁻¹ G418.

### Cloning of DarT-TRF1

FlpIn-GFP-DarT-TRF1 WT and E160A were generated by cloning fragments of pEGFPC1 N-GFP-DarT WT/E160A and TRF1 from ER-DD-mCherry TRF1-FokI into pFRT/lacZeo using the In-Fusion HD Cloning Plus (Takara Bio). Plasmids were confirmed via sequencing.

### Generation of DarT-TRF1 cell lines

CTRL and TARG1-KO U2OS FlpIn cell lines were engineered using the Synthego CRISPR editing kit with RNPs. Both the FlpIn-DarT-TRF1 and pOG44 (Flp recombinase) vectors were transfected to create doxycycline-inducible DarT-TRF1 WT and E160A, U2OS CTRL and TARG1-KO cell lines. These cell lines were confirmed by western blot.

### siRNA transfections

Briefly, 200,000 and 700,000 cells were seeded per well of a six-well plate and 10 cm dish containing growth medium without antibiotics. Roughly 6 h later, cells were transfected. Small-interfering RNAs (siRNAs) and Dharmafect were diluted in OptiMEM (Life Technologies). A working siRNA concentration of 50 nM was used. We used 2.5 or 5 µl of siRNA and 10 or 25 µl of Dharmafect per well per 10 cm plate. Transfection medium was replaced with complete culture media -16 h later, or cells were split for the desired application and gathered at 72 h post-transfection. The following siRNA Smartpools or individual sequences from Dharmacon were used:

NT: D-001810; TARG1: L-015886; ARH3: L-020822; HPF1: L-020849; PARP1: L-006656;

PARP2: L-010127; PARG: L-011488; ERCC2: L-011027; ERCC3: L-011028; POT1: L-004205; Lig1: L-011076-00; FEN1: L-010344-00.

### Small molecule inhibitors

The following small molecule inhibitors were used in this study: PARPi (Olaparib; Selleck Chemicals, catalog no. S1060), PARGi (PDD 00017273; Tocris Bioscience, catalog no. 5952), ATRi (AZD6738; Selleck Chemicals, catalog no. S7693) and FEN1i (MedChemExpress; catalog no. HY-136485).

### Clonogenic survival assays

For these, 2,000 U2OS cells were seeded in six-well plates in triplicate and cultured for 7 days before fixation and staining in a 1% crystal violet

solution. Plates were imaged with Typhoon 9400 PhosphoImager (GE Healthcare) and analyzed with Fiji, which was used to count positively stained colonies and calculate total cell coverage per well.

### Direct immunofluorescence

Cells on glass coverslips were washed twice in PBS and fixed with 2% paraformaldehyde (PFA) for 10 min. Cells were permeabilized with 0.1% (w/v) sodium citrate and 0.1% (v/v) Triton X100 for 5 min and incubated with fresh blocking solution (1 mg ml⁻¹ BSA, 10% normal goat serum, 0.1% Tween) for 30 min. Primary antibodies were diluted in blocking solution and added to cells for 1 h at room temperature or overnight refrigerated. Next, cells were washed three times with PBS for 5 min and incubated with Alexa-coupled secondary antibodies (488, 555, 647 nm) (Life Technologies) for 1 h at room temperature. Then, cells were washed three times with PBS and mounted on slides with Prolong Gold Anti-fade reagent with 4,6-diamidino-2-phenylindole (DAPI) (Life Technologies). Once the Prolong Anti-fade polymerized and cured, cells were visualized by conventional fluorescence with ×40 and ×63 Plan λ objectives (1.4 oil) using a Nikon 90I. For additional antibody information, see Supplementary Table 1.

### IF-FISH

After secondary antibody, cells were washed and then the immunofluorescence (IF) staining was fixed with 2% PFA for 10 min. PFA was washed off with PBS and coverslips dehydrated with successive washes in 70, 95 and 100% EtOH for 3 min, allowed to air dry completely. Next, the coverslips were mounted on glass slides with 15 ml per coverslip of hybridization mix (70% deionized formamide, 1 mg ml⁻¹ of Blocking Reagent (Roche), 10 mM Tris-HCl pH 7.4) containing Alexa 488-(CCCTAA)₄ PNA probe (PNA Bio). DNA was denatured by setting the slides on a heating block set to 72 °C for 10 min and then incubating overnight at room temperature in the dark. The coverslips were then washed twice for 15 min with Wash Solution A (70% deionized formamide and 10 mM Tris-HCl pH 7.2) and three times with solution B (0.1 M Tris-HCl pH 7.2, 0.15 M NaCl and 0.08% Tween) for 5 min at room temperature. EtOH dehydration was repeated as above, and finally, the samples were mounted and analyzed as mentioned.

### CO-FISH

CO-FISH was performed as described[48]. In brief, cell cultures were incubated with 7.5 mM BrdU and 2.5 mM BrdC for roughly 12 h. After removal of nucleotide analogs, Colcemid (Gibco) was added for around 2 h, cells were gathered by trypsinization, swelled in 75 mM KCl and fixed in 70% methanol:30% acetic acid. Samples were stored at −20 °C overnight. Metaphase chromosomes were spread by dropping onto washed slides, then RNaseA (0.5 mg ml⁻¹) and pepsin treated. Slides were incubated in 2× SSC containing 0.5 mg ml⁻¹ Hoechst 33258 for 15 min in the dark and irradiated for 40 min (5.4 × 10⁵ J m⁻², energy 5400) at in a ultraviolet (UV) Stratalinker 2400 (Stratagene). The nicked BrdU/C substituted DNA strands were degraded by Exonuclease III digestion. The slides were then washed in PBS, dehydrated by EtOH washes and allowed to air dry completely. The remaining strands were hybridized with fluorescence-labeled DNA probes of different colors, specific either for the positive telomere strand (TTAGGG)₄ (polymerized by lagging strand synthesis) (Alexa 488, green color), or the negative telomere strand (CCCTAA)₄ (polymerized by leading strand synthesis) (Alexa-568, red color). Before hybridization of the first PNA, DNA is denatured by heating at 72 °C for 10 min, as in immunofluorescence-FISH (see the section 'IF-FISH'), and then incubated for 2 h at room temperature. Slides were washed for 15 min with Wash Solution A (see the section 'IF-FISH'), dried and then incubated with the second PNA for 2 h at room temperature. The slides were then washed again twice for 15 min with Wash Solution A and three times with Wash Solution B (see the section 'IF-FISH') for 5 min at room temperature. The second wash contained DAPI (0.5 µg ml⁻¹). Finally, cells were dehydrated in EtOH as

above and mounted (Vectashield). The resulting chromosomes showed dual staining and allowed distinction between leading and lagging strands. Metaphase chromosomes were visualized by conventional fluorescence microscope with a ×63 Plan λ objective (1.4 oil) on a Nikon 90i microscope.

## Purification of leading and lagging telomeres

Purification of leading and lagging telomeres by CsCl centrifugation was performed as described[49]. Briefly, cells were incorporated with 100 µM IdU for 20 h. Five hundred micrograms of genomic DNA was purified using Puregene Core Kit A (Qiagen) and then digested with HinfI/RsaI. DNAs were mixed with CsCl solution to obtain a final density of 1.770 g ml$^{-1}$. Samples were centrifuged at 240,241.6$g$ for 20 h using a VTi-90 vertical rotor (Beckman) and then fractions were collected. Slot blot for each fraction was performed to collect the leading and lagging strand fractions. Leading DNA was located at a density of 1.790–1.800 g ml$^{-1}$; lagging DNA was located at a density of 1.760–1.770 g ml$^{-1}$ and unreplicated DNA was located at a density of 1.740–1.750 g ml$^{-1}$.

## SMAT

SMAT replication was performed as described[50]. Briefly, cells were labeled with 10 µM EdU for 5 h before collection by trypsinization. Cells were embedded in low-melting agarose plugs and then subjected to proteinase K digestion overnight. Plugs were dissolved with agarose (Thermo Scientific) according to the manufacturer's instructions. Molecular combing was performed using the Molecular Combing System (Genomic Vision S.A.) with a constant stretch factor of 2 kb per µm using vinyl silane coverslips (20 × 20 mm; Genomic Vision S.A.), according to the manufacturer's instructions. After combing, coverslips were dried for 3 h at 60 °C. The quality and integrity of combed DNA fibers were checked using the YoYo-1 counterstain (Molecular Probes). Coverslips were stained for telomeres, EdU-label and DNA fibers via YoYo-1 counterstain, as previously described[4]. In brief, telomeric DNA was visualized by hybridization with a TAMRA–OO-(TTAGGG)$_3$ PNA probe (Panagene) followed by a 5 min wash with 50% formamide and 1× SSC, 2 × 5 min washes with 2× SSC and 1 × 5 min wash medium shaking with 2× SSC and 0.1% Tween. Following PNA probe hybridization, samples were fixed for 10 min (4% formaldehyde) and Click-it detection of EdU was applied simultaneously with YoYo-1 staining. Telomere fibers were detected on a Zeiss Axio Imager microscope using the ×63 oil objective and analyzed manually with Zen software (Zeiss).

## Purification of telomere DNA

Region-specific DNA purification was conducted as described[19]. Approximately 10$^7$ cells per condition were collected by scraping in PBS containing 1 µM PARPi and 1 µM PARGi on ice. Cells were washed with 1× PBS and centrifuged (500$g$, 5 min, 4 °C). Cell pellets were resuspended in 350 µl of TNE (10 mM Tris pH 7.4, 100 mM NaCl, 10 mM EDTA) and 350 µl of TNES:proteinase K (10 mM Tris pH 7.4, 100 mM NaCl, 10 mM EDTA, 1% SDS, 100 µg ml$^{-1}$ proteinase K) overnight at 37 °C. Genomic DNA was then extracted using phenol:chloroform. Then 250 µg of DNA was incubated with *HinFI*, *HphI*, *MnlI* and RE buffer (330 mM Tris-HCl, pH 8.0, 100 mM magnesium acetate, 1 M LiCl, 5 mM dithiothreitol) overnight at 37 °C. The digested DNA was then combined with RSE-H buffer (polymerase and deoxyribonucleotide triphosphates) and the telomere capture oligo (Biotin-5′-ATCC(CCCTAA)$_3$-3′). The samples were then placed at 64 °C for 20 min for denaturation and primer extension. The RSE reaction was then combined with 90 µl of RSE-B beads and incubated for 1 h rotating at room temperature. The beads were then subjected to two washes of 500 µl of RSE Wash Buffer for 5 min each rotating at room temperature. Beads were resuspended in 45 µl of RSE Resuspension Buffer R and eluted in an 82 °C water bath for 5 min, then left to sit overnight to come to room temperature. DNA was then quantified and prepped for dot blotting.

## Dot blot detection of ADPr

Samples were dot-blotted onto a positively charged nylon membrane for detection. Samples were cross-linked to the membrane (UV Stratalinker 2400, Stratagene). The membrane was blocked in 5% milk in TBST (10 mM Tris-HCl, 15 mM NaCl, 0.05% Tween-20) at room temperature for 30 min and incubated with primary antibody at 4 °C overnight. The next day, the membrane was washed three times in TBST. Horseradish peroxidase (HRP)-linked antirabbit or mouse (Amersham) was used for secondary antibodies, and the HRP signal was visualized with SuperSignal ECL (enhanced chemiluminescence) substrate (Pierce) per the manufacturer's instructions. To detect telomeric DNA, samples were similarly dot-blotted onto a positively charged nylon membrane. Samples were denatured (0.5 M NaOH and 1.5 M NaCl) for 15 min and neutralized (0.5 M Tris-HCL pH 7.5 and 1 M NaCl) for 10 min. Samples were cross-linked to the membrane (UV Stratalinker 2400, Stratagene). The membrane was prehybridized in a hybridization buffer (Ultrahyb Ultra-sensitive Hybridization Buffer, Invitrogen) rotating at 55 °C for 30 min. The membrane was hybridized with $^{32}$P-labeled (AATCCC)$_4$ oligonucleotides at 55 °C overnight. The next day, the membrane was washed three times in 2× SSC buffer and once in 2× SSC 0.5% SDS, exposed onto a storage phosphor screen and scanned using Typhoon 9400 PhosphoImager (GE Healthcare). Sample intensity was measured with Fiji. For additional antibody information, see Supplementary Table 1.

## In vitro hydrolase assays

For testing the removal of ADP-ribosylation modifications from telomeric DNA by ADP-ribosyl-hydrolases, purified telomeric DNA samples were incubated with buffer as control or 1 µM of the indicated hydrolase at 37 °C for 30 min in assay buffer consisting of 50 mM TRIS pH 8.0, 50 mM NaCl, 1 mM MgCl$_2$. Reaction products were analyzed by dot blot assay. For this, 15 ng of DNA was dotted onto a nitrocellulose membrane (Amersham Protran 0.45 NC nitrocellulose) and cross-linked with 1200J using a Stratalinker UV crosslinker. Cross-linked DNA was then immunoblotted for ADPr-DNA using the poly/mono-ADP-ribose, antibody (below) for 1 h at room temperature in 5% (w/v) powdered milk in PBS-T. Secondary peroxidase-couple antibodies (Dako-Agilent) were incubated at room temperature for 1 h. ECL-based chemiluminescence was detected using Pierce ECL Western Blotting Substrate (Thermo Scientific) and Hyperfilms (GE).

## Western blotting

Cells were collected with trypsin, quickly washed in PBS, counted with Cellometer Auto T4 (Nexcelom Bioscience) and directly lysed in 4× NuPage LDS sample buffer at 10,000 cells per µl. PARPi and PARGi were added to 4× NuPage LDS sample buffer to inhibit PARP and PARG activity in solution, except for the westerns shown in Fig. 3d. Proteins were gently homogenized using a Benzonase (ThermoFisher), denatured for 10 min at 70 °C and resolved by SDS–PAGE electrophoresis, transferred to nitrocellulose membranes, blocked in 5% milk in TBST for 30 min and probed. HRP-linked antirabbit or mouse (Amersham) was used for secondary antibodies, and the HRP signal was visualized with SuperSignal ECL substrate (Pierce) as per the manufacturer's instructions. For additional antibody information, see Supplementary Table 1.

## TRF analysis by pulse field gel electrophoresis

Telomere gels were performed using telomere restriction fragment (TRF) analysis. Genomic DNA was digested using AluI and MboI (NEB). Next, 4–10 µg of DNA was run on a 1% PFGE agarose gel (Bio-Rad) in 0.5× TBE buffer using the CHEF-DRII system (Bio-Rad) at 6 V cm$^{-1}$; initial switch time 1 s, final switch time 6 s, for 17 h at 14 °C. The gel was then dried for 2 h at 60 °C, denatured in a 0.5 N NaOH 1.5 M NaCl solution and neutralized. The gel was hybridized with $^{32}$P-labeled (TTAGGG)$_4$ oligonucleotides in Church buffer overnight at 55 °C. The next day, the membrane was washed three times in 2× SSC buffer and once in 2× SSC

0.5% SDS, exposed onto a storage phosphor screen and scanned using Typhoon 9400 PhosphoImager (GE Healthcare).

## Quantification and statistical analysis
All data in this study were analyzed in GraphPad Prism, ImageJ and Microsoft Excel. Detection, colocalization and quantification of cells were performed using the ComDet v.0.5.3 plugin for ImageJ. Statistical tests used are indicated in the figure legend accompanying each figure. In most cases, one-way analysis of variance (ANOVA) was used. Typically, unless otherwise stated, *n* refers to the number of independent experiments and s.e.m. refers to the standard error of means. The sample size was not predetermined.

## Reporting summary
Further information on research design is available in the Nature Portfolio Reporting Summary linked to this article.

## Data availability
All data are available in the main, Extended Data and Supplementary Information figures. Original data are deposited at figshare https://doi.org/10.6084/m9.figshare.25103732 (ref. 51) or available from the corresponding author upon request. Source data are provided with this paper.

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

## Acknowledgements
We thank I. Matic (Max Planck Institute for Aging, Köln) for providing mono-ADPr-specific antibodies used in this study. We are grateful to J. Nandakumar, K. Brenner (University of Michigan), G. Glousker and J. Lingner (EPFL, Switzerland) for POT1-KO and POT1 complemented HEK293E cells. We thank M. De Rosa for providing lentivirus and A. Detwiler for technical assistance with RSE. We thank all authors and members of the O'Sullivan and Ahel laboratories for critically reading the manuscript. The following grants were received by: R.J.O., grant no. R01CA207209 (NCI), grant no. R01CA262316 (NCI) and CCSG grant no. P30CA047904 (NCI); I.A., Welcome Trust (grant nos. 101794 and 210634), Ovarian Cancer Research Alliance (Collaborative Research Development grant no. 813369) and Cancer Research UK (C35050/A22284); A.R.W., National Institutes of Health training award T32 grant no. GM133332 (Department of P&CB, University of Pittsburgh) and a Ruth L. Kirschstein National Research Scientist Training Award, F30 grant no. CA278287; P.L.O., grant no. R35ES030396 (NIEHS); J.M., grant no. K22CA245259 (NCI) and H.A.P., grant no. 2007488 (Medical Research Future Fund, Australia). Work in the Ahel Laboratory is supported by the Biotechnology and Biological Sciences Research Council (grant nos. BB/R007195/1 and BB/W016613/1), the Wellcome Trust (grant nos. 210634 and 223107) and Cancer Research UK (grant no. C35050/A22284). The funders had no role in study design, data collection and analysis, decision to publish or preparation of the manuscript.

## Author contributions
Conceptualization was carried out by A.R.W., I.A and R.J.O. The methodology was developed by A.R.W., J.L., R.L., R.B., L.C.C., M.S., J.G., S.S.-H. and P.L.O. The original draft was written by A.R.W. and R.J.O. Funding was acquired by J.M., H.A.P., P.L.O., I.A. and R.J.O. The work was supervised by J.M., H.A.P., I.A. and R.J.O. Project administration was the responsibility of R.J.O.

## Competing interests
The authors declare no competing interests.

## Additional information
**Extended data** is available for this paper at https://doi.org/10.1038/s41594-024-01279-6.

**Correspondence and requests for materials** should be addressed to Roderick J. O'Sullivan.

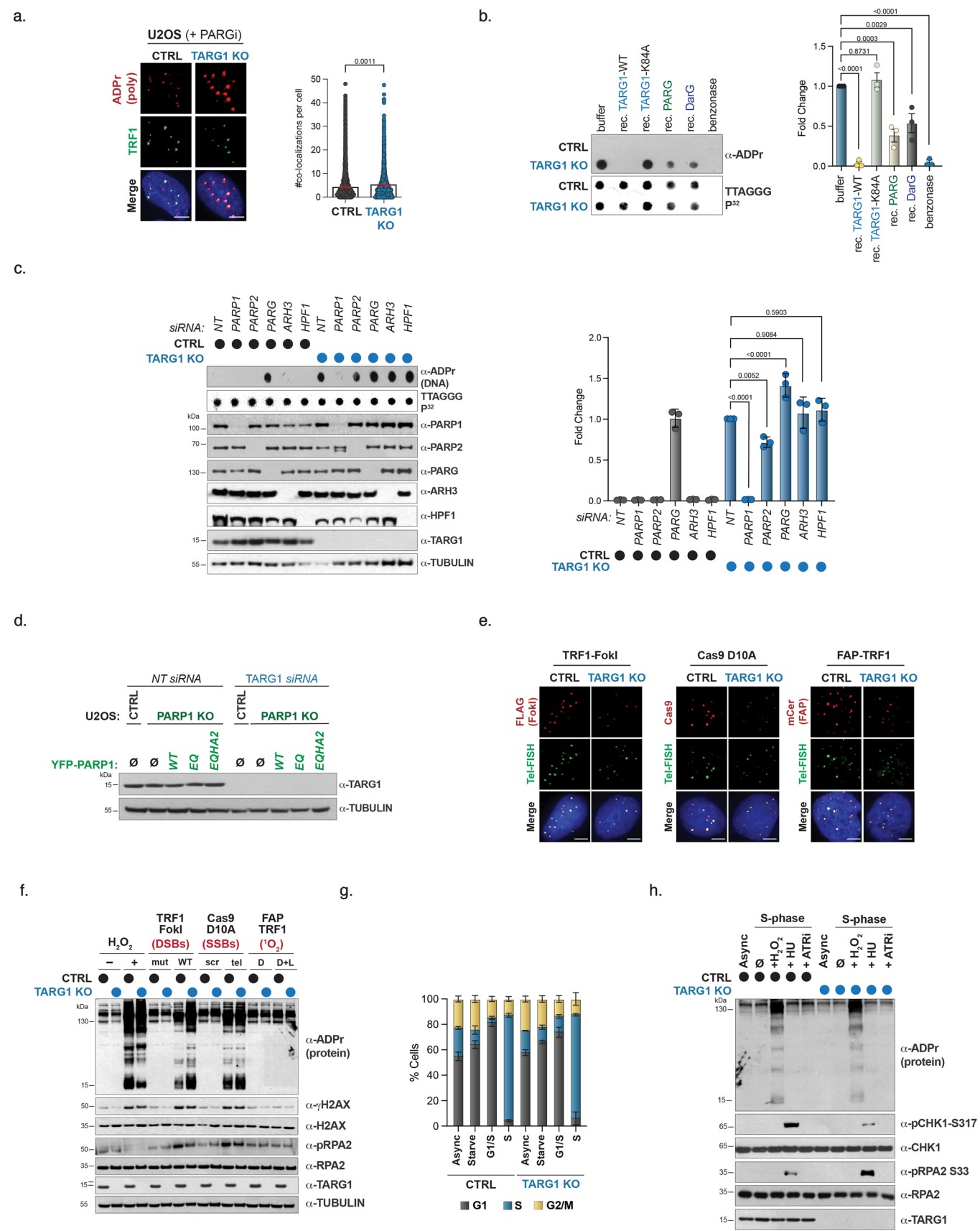

Extended Data Fig. 1 | See next page for caption.

**Extended Data Fig. 1 | DNA-ADPr is stimulated by PARP1 and removed by TARG1. a**) IF of poly ADP-ribose U2OS CTRL and TARG1 KO cells with PARGi treatment (96 hours). Images (left) and quantification of the number of ADPr+ telomeres (TRF1) per cell (right). Data represent the mean and s.e.m. of biological replicates (>4000 cells counted per condition). A two-tailed Student's $t$-test was performed on n = 2 biologically independent experiments. **b**) Telomeric western dot blot of ADPr and Southern blot of telomeric DNA of U2OS CTRL and TARG1 KO telomeric DNA treated with full-length TARG1, K84A TARG1, PARG, and DarG proteins (left) and quantification (right). **c**) Telomeric western dot blot of ADPr and Southern blot of telomeric DNA and western blot (left) and ADPr quantification (right) of U2OS CTRL and TARG1 KO after control, PARP1, PARP2, PARG, ARH3, or HPF1 knockdown normalized to TARG1 KO NT siRNA. **d**) Western blot of U2OS CTRL, PARP1 KO, and PARP1 KO with expression of either YFP-tagged full-length (WT) PARP1, PARP1 EQ, or PARP1-EQHA2 after control

or TARG1 knockdown. **e**) IF representative images of expression of TRF1-FokI (FLAG-red), Cas9 D10A (Cas9-red), or FAP-TRF1 (mCer-red) and telomere FISH (green) in U2OS CTRL and TARG1 KO cells. This was repeated three biologically independent times with similar results. **f**) Western blot of U2OS CTRL and TARG1 KO cells after treatment with $H_2O_2$ or expression of TRF1-FokI (D450A or WT), Cas9 D10A (scr or tel), or FAP-TRF1 (dye or dye and light). **g**) Flow cytometry of U2OS CTRL and TARG1 KO cells in asynchronous (async), serum-starved (starve), G1/S, or S phases. Mean and s.e.m. are shown for three independent experiments for G1 (grey), S (blue), and G2/M (yellow). **h**) Western blot of U2OS CTRL and TARG1 KO in either asynchronous (async) or S phase treated with $H_2O_2$ (2 mM–15 mins), HU (2mM- 1 hr), or ATRi (5 nM- 1 hr). Mean and s.e.m. are shown from three independent experiments in (B) and (C), and groups were compared with a one-way ANOVA followed by Tukey's multiple-comparisons test for pairwise comparisons. For (A) and (E), scale bars represent 10 um.

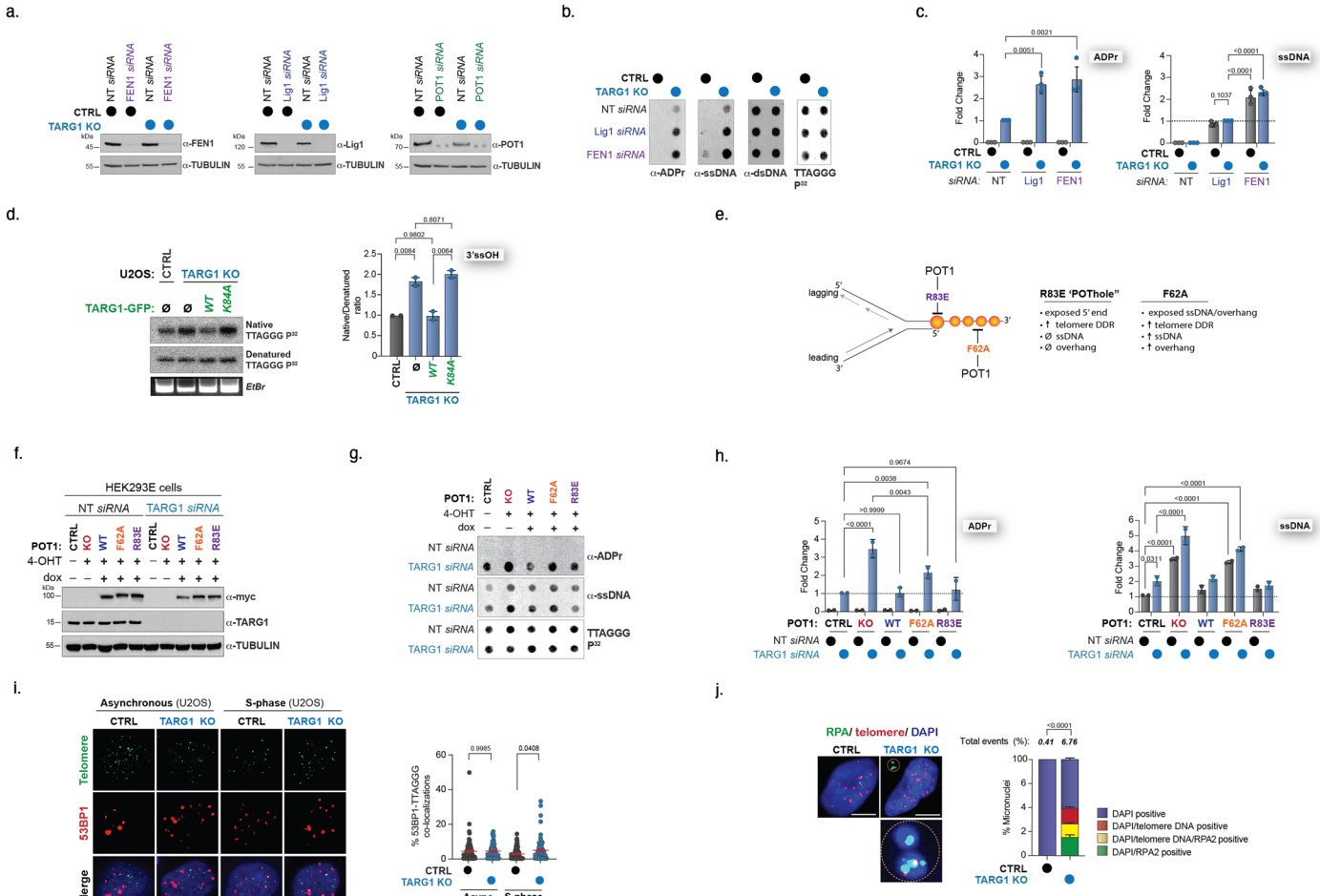

**Extended Data Fig. 2 | TARG1 deficiency impairs telomere replication.**
**a**) Western blots of U2OS CTRL and TARG1 KO cells after knockdown of FEN1 (left), Lig1 (middle), or POT1 (right). **b**) Telomeric western dot blot of ADPr and Southern blot of telomeric DNA of U2OS CTRL and TARG1 KO after control, Lig1 or FEN1 knockdown. **c**) Quantification ADPr and ssDNA from (b) normalized to TARG KO NT siRNA. **d**) GFP-tagged full-length (WT) TARG1 or TARG1-K84A DNA ran on a gel and probed for telomeric DNA in native and denaturing Southern blots. Quantification of the ratio of native/denatured telomeric DNA detected normalized to CTRL (right). Mean and s.e.m. are shown from n = 2 biologically independent experiments and groups were compared with a one-way ANOVA followed by Tukey's multiple-comparisons test for pairwise comparisons. **e**) Schematic of separation of function POT1 mutants. The R83E mutant exposes the 5' end at the double/single-stranded DNA junction and the F62A mutant increases the exposed ssDNA. **f**) Western blot of HEK293E POT1 cells treated with OHT (0.5 uM) to deplete POT1 and then doxycycline (1 ug/mL) to express either myc-tagged WT POT1, F62A POT1, or R83E POT1 with control or TARG1

knockdown. **g**) Telomeric western dot blot of ADPr and ssDNA and Southern blot of telomeric DNA with conditions used in (e). **h**) Quantification of ADPr and ssDNA from (f) normalized to CTRL TARG1 siRNA and CTRL NT siRNA, respectively. **i**) IF-FISH of 53BP1 and telomeres in U2OS CTRL and TARG1 KO cells. Images (left) and quantification of ADPr+ telomeres (right). Data represent the mean and s.e.m. of n = 2 biological replicates ( > 100 cells analyzed per condition) compared with one-way ANOVA followed by Tukey's multiple-comparisons test for pairwise comparisons. **j**) IF-FISH of RPA2 and telomeres in U2OS CTRL and TARG1 KO cells. Yellow circles highlight micronuclei (MN) containing RPA2 foci and TTAGGG signals. Images (left) and quantification of DAPI-positive, DAPI-telomere, and DAPI-RPA2-telomere-positive micronuclei. Data represent the mean and s.e.m. n = 3 biological replicates ( > 1000 cells analyzed (Two-tailed student's *t*-test)). Mean and s.e.m. are shown from three independent experiments in (C) and (H), and groups were compared with a one-way ANOVA followed by Tukey's multiple-comparisons test for pairwise comparisons. For (I) and (J), scale bars represent 10 um.

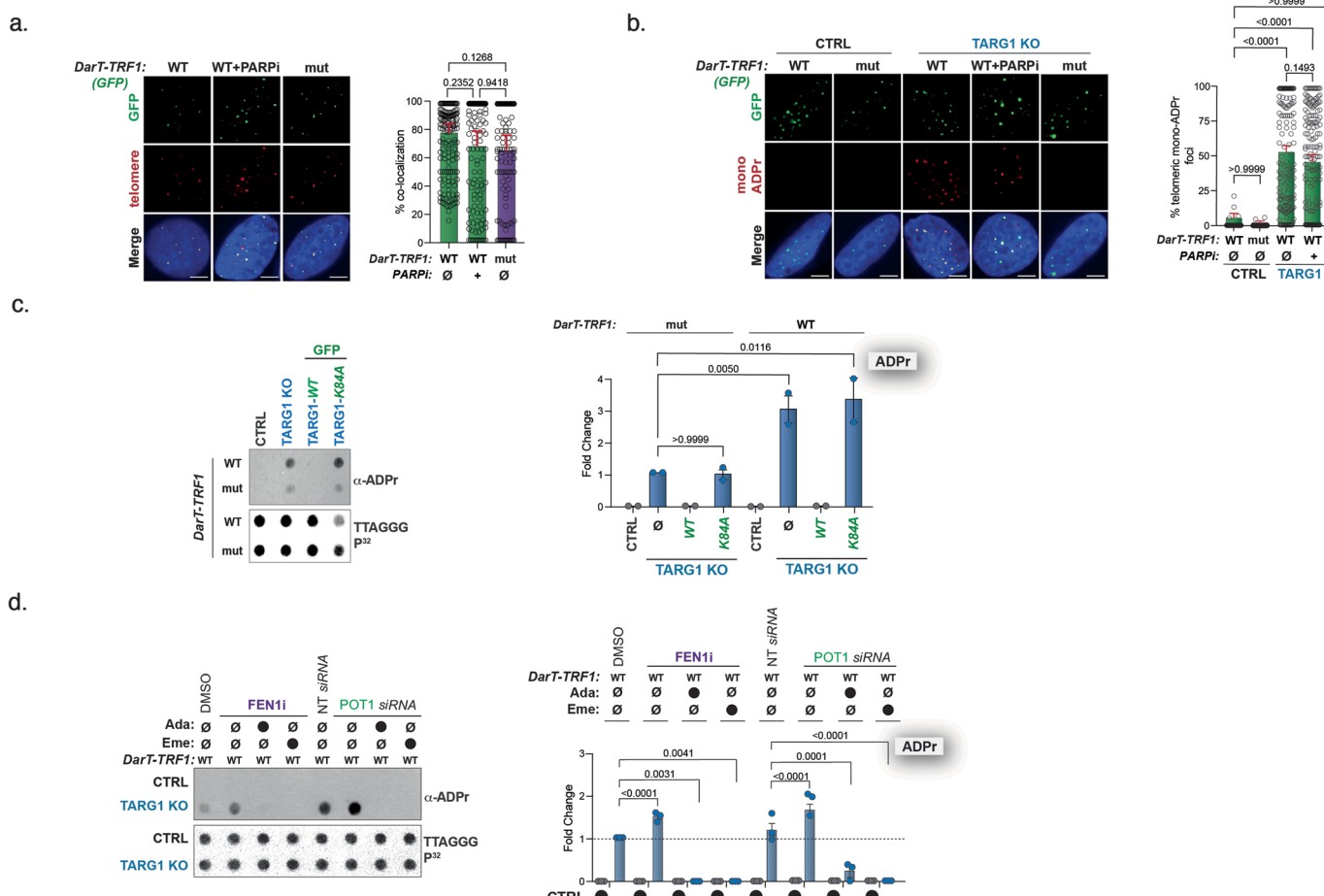

**Extended Data Fig. 3 | DarT-TRF1 targets DNA-ADPr to the telomere. a)** IF images (left) of WT DarT-TRF1 (WT) treated with PARPi and mut DarT-TRF1 (mut) and quantification of colocalization of the telomere (TRF2) and GFP (right) with more than 100 cells analyzed per condition in n = 2 biologically independent experiments. **b)** IF images (left) of U2OS CTRL and TARG1 KO cells after transient expression of WT DarT-TRF1 (WT) treated with PARPi and mut DarT-TRF1 (mut) and quantification of the colocalization of GFP (telomere) and mono-ADPr (red) (right) with more than 150 cells analyzed per condition. N = 3 shown from three biologically independent experiments. **c)** Telomeric western dot blot of ADPr and Southern blot of telomeric DNA of U2OS CTRL and TARG1 KO with reconstitution of either GFP-tagged full-length (WT) TARG1 or TARG1-K84A after transient expression of WT DarT-TRF1 (WT) or mut DarT-TRF1 (mut) and quantification

ADPr (right) normalized to TARG1 KO with mut DarT-TRF1 expression. N = 2 shown from two biologically independent experiments. **d)** Telomeric western dot blot (left) Southern blot of telomeric DNA of U2OS CTRL and TARG1 KO cells after expression of WT DarT-TRF1 and either FEN1 inhibition (FEN1i) or POT1 knockdown (siPOT1) treated with adarotene (ADA) or emetine (EME) and quantification of ADPr (right) normalized to TARG1KO with WT DarT-TRF1 expression treated with DMSO. N = 3 shown from three biologically independent experiments. Mean and s.e.m. are shown from at least two independent experiments in (A), (B), (C) and (D), and groups were compared with a one-way ANOVA followed by Tukey's multiple-comparisons test for pairwise comparisons. For (A) and (B), scale bars represent 10 um.

**a.**

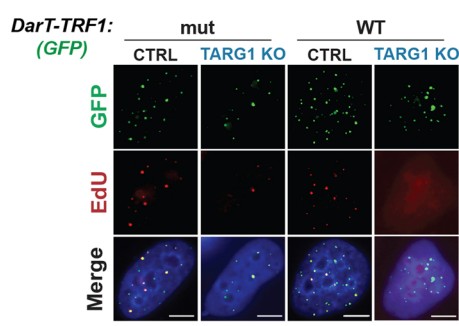

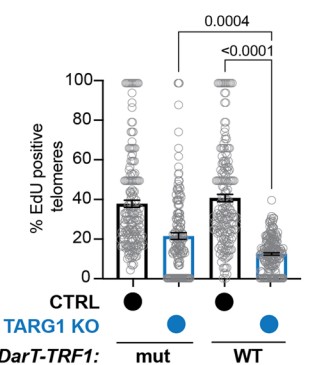

**b.**

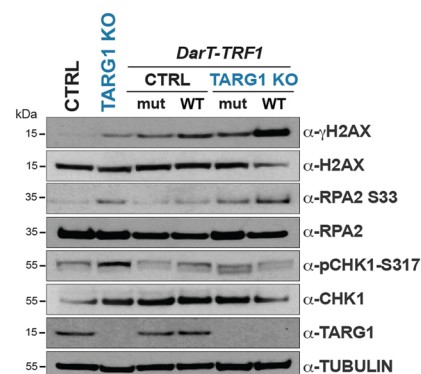

**c.**

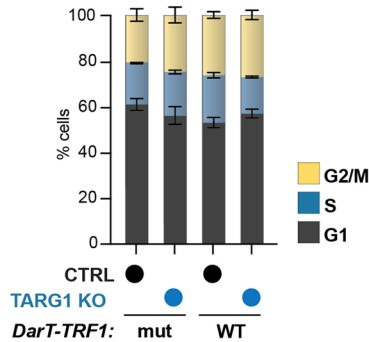

**d.**

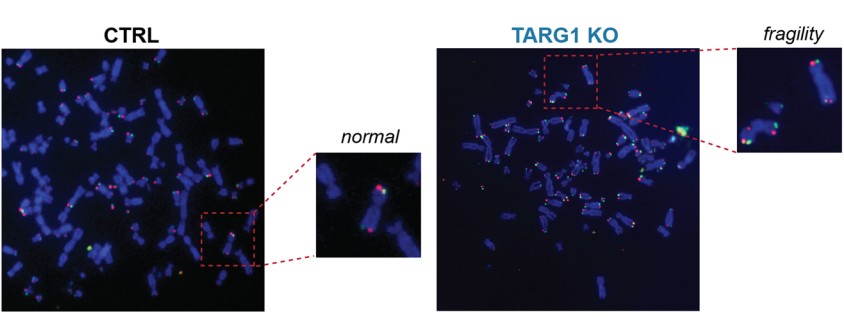

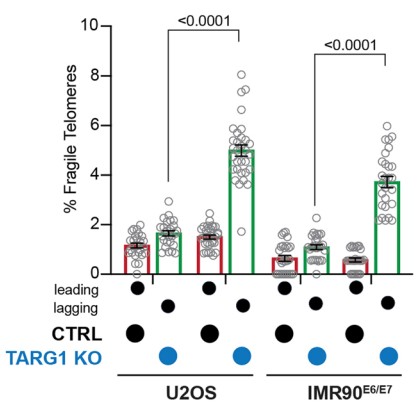

**e.**

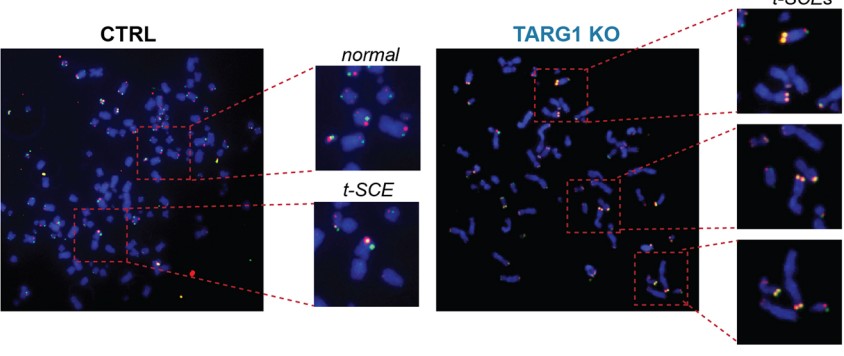

**Extended Data Fig. 4 | See next page for caption.**

**Extended Data Fig. 4 | TARG1 loss impairs telomere integrity. a**) IF images (left) of EdU-pulsed U2OS CTRL and TARG1 KO cells after expression of WT DarT-TRF1 (WT) or E160A DarT-TRF1 (mut) and quantification of % EdU-positive telomeres (right). ( > 150 cells analyzed per condition). Mean and s.e.m. are shown from three independent experiments and groups were compared with a one-way ANOVA followed by Tukey's multiple-comparisons test for pairwise comparisons. Scale bars represent 10 um. **b**) Western blot of U2OS CTRL and TARG1 KO cells after transient expression of WT DarT-TRF1 (WT) or mut DarT-TRF1 (mut). **c**) Flow cytometry of U2OS CTRL and TARG1 KO cells after inducible expression of WT DarT-TRF1 (WT) or mut DarT-TRF1 (mut). Mean and s.e.m. of three biologically independent experiments are shown for G1 (grey), S (blue), and G2/M (yellow). **d**) Chromosome-Orientation FISH (CO-FISH) representative images of normal and lagging (green) strand fragile telomeres of U2OS CTRL and TARG1 KO cells and quantification of % leading (red) and lagging (green) fragile telomeres in U2OS and IMR90$^{E6/E7}$ CTRL AND TARG1 KO cells (>7000 U2OS and >5000 IMR90$^{E6/E7}$ telomeres scored). **e**) CO-FISH representative images of normal telomeres and telomere sister chromatid exchanges (telomere SCEs) of U2OS CTRL and TARG1 KO cells and quantification of % telomere SCEs from cells in (D). Mean and s.e.m. are shown from three independent experiments in (D) and (E) and groups were compared with a two-tailed Student's $t$-test.

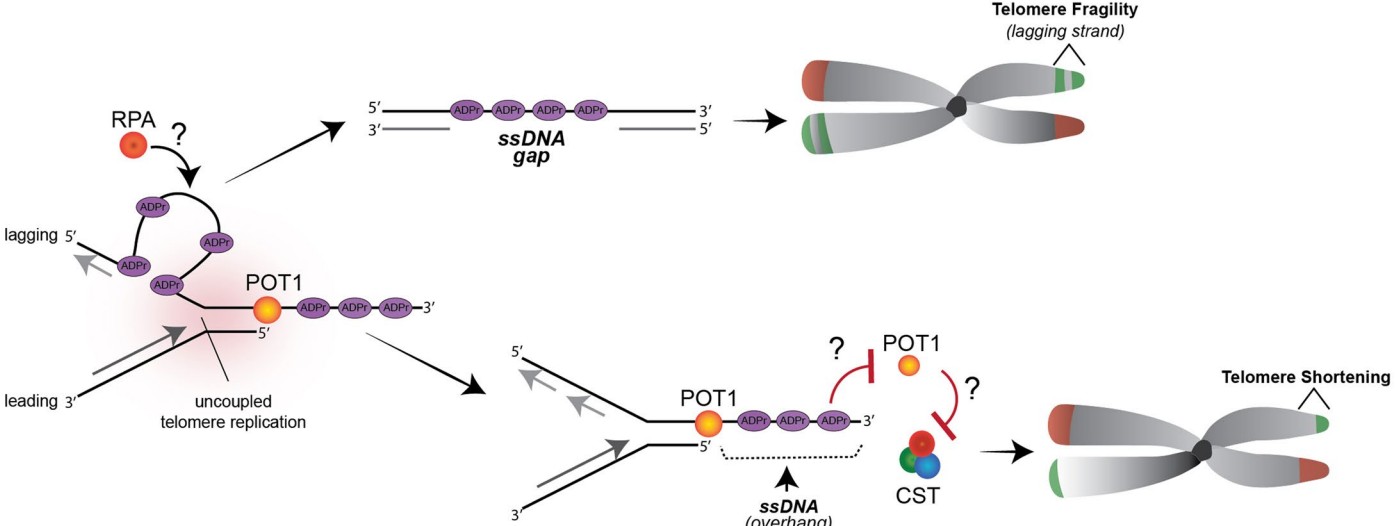

**Extended Data Fig. 5 | Model for how persistent DNA-ADPr impacts telomere replication.** During telomere replication in S-phase, single-stranded DNA at unligated Okazaki fragments on the lagging strand stimulates DNA ADP-ribosylation (DNA-ADPr) by PARP1. Normally, this DNA-ADPr is quickly removed by TARG1's hydrolytic activity to maintain DNA replication fidelity and progression. However, in TARG1-deficient cells, the unhydrolyzed DNA-ADPr interferes with telomere replication leading to the uncoupling of DNA strand synthesis and persistent post-replicative DNA-ADPr ssDNA gaps in telomeres that are associated with the fragile telomere phenotype. In addition, the availability of ssDNA on the 3'overhang might also attract PARP1's DNA-ADPr activity. Failure to remove DNA-ADPr from the 3'overhang might interfere with telomere replication by briefly preventing POT1 binding and/or by disrupting fill-in DNA synthesis that is mediated by the CTC1-STN1-TEN1 (CST)-Polα primase complex, leading to telomere shortening in TARG1 deficient cells.

# Reporting Summary

## Statistics

For all statistical analyses, confirm that the following items are present in the figure legend, table legend, main text, or Methods section.

| n/a | Confirmed | |
|---|---|---|
| ☐ | ☒ | The exact sample size (*n*) for each experimental group/condition, given as a discrete number and unit of measurement |
| ☐ | ☒ | A statement on whether measurements were taken from distinct samples or whether the same sample was measured repeatedly |
| ☐ | ☒ | The statistical test(s) used AND whether they are one- or two-sided *Only common tests should be described solely by name; describe more complex techniques in the Methods section.* |
| ☒ | ☐ | A description of all covariates tested |
| ☐ | ☒ | A description of any assumptions or corrections, such as tests of normality and adjustment for multiple comparisons |
| ☐ | ☒ | A full description of the statistical parameters including central tendency (e.g. means) or other basic estimates (e.g. regression coefficient) AND variation (e.g. standard deviation) or associated estimates of uncertainty (e.g. confidence intervals) |
| ☐ | ☒ | For null hypothesis testing, the test statistic (e.g. *F*, *t*, *r*) with confidence intervals, effect sizes, degrees of freedom and *P* value noted *Give P values as exact values whenever suitable.* |
| ☒ | ☐ | For Bayesian analysis, information on the choice of priors and Markov chain Monte Carlo settings |
| ☒ | ☐ | For hierarchical and complex designs, identification of the appropriate level for tests and full reporting of outcomes |
| ☒ | ☐ | Estimates of effect sizes (e.g. Cohen's *d*, Pearson's *r*), indicating how they were calculated |

*Our web collection on statistics for biologists contains articles on many of the points above.*

## Software and code

Policy information about availability of computer code

| Data collection | Nikon 90I Microscope, NIS Elements |
|---|---|
| Data analysis | GraphPad Prism (10.0.0), ImageJ2 (Version: 2.14.0/1.54f), and Microsoft Excel (16.66), ComDet v.0.5.3 plugin for ImageJ, Zen |

For manuscripts utilizing custom algorithms or software that are central to the research but not yet described in published literature, software must be made available to editors and reviewers. We strongly encourage code deposition in a community repository (e.g. GitHub). See the Nature Portfolio guidelines for submitting code & software for further information.

## Data

Policy information about availability of data

All manuscripts must include a data availability statement. This statement should provide the following information, where applicable:
- Accession codes, unique identifiers, or web links for publicly available datasets
- A description of any restrictions on data availability
- For clinical datasets or third party data, please ensure that the statement adheres to our policy

All data are available in the main, extended data, and supplementary data figures. Original data are deposited at Figshare doi:10.6084/m9.figshare.25103732 or available from the corresponding author upon request.

# Research involving human participants, their data, or biological material

Policy information about studies with human participants or human data. See also policy information about sex, gender (identity/presentation), and sexual orientation and race, ethnicity and racism.

| | |
|---|---|
| Reporting on sex and gender | N/A |
| Reporting on race, ethnicity, or other socially relevant groupings | N/A |
| Population characteristics | N/A |
| Recruitment | N/A |
| Ethics oversight | N/A |

Note that full information on the approval of the study protocol must also be provided in the manuscript.

# Field-specific reporting

Please select the one below that is the best fit for your research. If you are not sure, read the appropriate sections before making your selection.

☒ Life sciences ☐ Behavioural & social sciences ☐ Ecological, evolutionary & environmental sciences

For a reference copy of the document with all sections, see nature.com/documents/nr-reporting-summary-flat.pdf

# Life sciences study design

All studies must disclose on these points even when the disclosure is negative.

| | |
|---|---|
| Sample size | No statistical method was used to predetermine sample size. Sample size was determined based on previous experiments. For metaphases, n=20-30 metaphases over 3 independent experiments for telomere fragility and TSCEs. |
| Data exclusions | No data were excluded from the analyses. |
| Replication | The number of biologically independent experiments performed is indicated in the legends. Experiments are representative of at least n=3 independent experiments unless indicated. All attempt at replication were found to be successful. Several experiments independently conducted by different researchers.Samples prepared in Pittsburgh were shipped to Oxford and independently assayed (Extended Data 1B) - this provided additional confirmation of TARG1 hydrolytic activity against DNA-ADPr.<br>Cell lines were shipped to dr PIckette annd Mins laboratories where experiments on ADPr at telomeres were corroborated |
| Randomization | Randomization was done when possible. Nuclei and metaphase imaging for all treatments were taken randomly. |
| Blinding | Investigators were not blinded except for metaphases experiments where investigators were blinded to the samples during imaging and analysis. All samples were treated identically for all experiments. |

# Reporting for specific materials, systems and methods

We require information from authors about some types of materials, experimental systems and methods used in many studies. Here, indicate whether each material, system or method listed is relevant to your study. If you are not sure if a list item applies to your research, read the appropriate section before selecting a response.

## Materials & experimental systems

| n/a | Involved in the study |
|---|---|
| ☐ | ☒ Antibodies |
| ☐ | ☒ Eukaryotic cell lines |
| ☒ | ☐ Palaeontology and archaeology |
| ☒ | ☐ Animals and other organisms |
| ☒ | ☐ Clinical data |
| ☒ | ☐ Dual use research of concern |
| ☒ | ☐ Plants |

## Methods

| n/a | Involved in the study |
|---|---|
| ☒ | ☐ ChIP-seq |
| ☒ | ☐ Flow cytometry |
| ☒ | ☐ MRI-based neuroimaging |

# Antibodies

| | |
|---|---|
| Antibodies used | Immunofluorescence: FLAG (D6W5B) (#14793S, Cell Signaling Technology, 1:1000); FLAG (M2) (#F1804, Millipore, 1:1000); TRF1 (#ab10579, Abcam, 1:1000); TRF2 (#NB110-57130, Novus, 1:1000); RPA2 (#ab2175 Lot #GR3386072-1, Abcam, 1:1000), Poly/Mono-ADP-ribose (#83732, Lot #5, Cell Signaling Technology, 1:1000); Mono-ADP-ribose (#HCA354 Lot #162452, Biorad, 1:1000); Mono-ADP-ribose (#HCA355 Lot #158442, Biorad, 1:1000); Cas9 (Cat. #14697 Lot #8, Cell Signaling Technology, 1:1000); GFP #ab6556, Abcam, 1:1000); Alexa Fluor 594 goat anti-rabbit (#A11012 Lot #2433881, Thermo Fisher, 1:1000); Alexa Fluor 594 goat anti-mouse. (#A11032 Lot #2527968, Thermo Fisher, 1:1000); Alexa Fluor 488 goat anti-mouse (#A11001 Lot #2659299, Thermo Fisher, 1:1000); Alexa Fluor 488 goat anti-rabbit. (#A32731 Lot #XD343356, Thermo Fisher, 1:1000); Alexa Fluor 647 goat anti-mouse. (#A21244 Lot #2527970, Thermo Fisher, 1:1000). <br><br> Dot blot: Pan-ADP-ribose (#MABE1016, Millipore, 1:1000); Poly/Mono-ADP-ribose (#83732, Lot #5, Cell Signaling Technology, 1:1000); Mono-ADP-ribose (#HCA354 Lot #162452, Biorad, 1:1000); Auto anti-dsDNA (#AB_10805293, University of Iowa, Developmental Studies Hybridoma Bank, 1:200); Auto anti-ssDNA #AB_10805144, University of Iowa, Developmental Studies Hybridoma Bank, 1:200). <br><br> Western blot: pRPA32 (S33) (#A300-246A, Lot #12, Bethyl, 1:1000); pRPA32 (S4/S8) (#A300-245A, Lot #9, Bethyl, 1:1000); pChk1 (S317) (#2344S, Lot #12, Cell Signaling Technology, 1:1000); Chk1 (#37010S, Lot #1, Cell Signaling Technology, 1:1000); gH2AX (#2577S Lot #12, Cell Signaling Technology, 1:1000); H2AX (#2595S Lot #8, Cell Signaling Technology, 1:1000); GFP (#130-091-833 Lot #5240106116, Miltenyi Biotech, 1:5000); PARG (#66564 Lot #1, Cell Signaling Technology, 1:1000); PARP1 (#61639 Lot #22922002, Active Motif, 1:1000); PARP2 (#39044 Lot #23822008, Active Motif, 1:1000); TARG1 (#25249-1-AP Lot #00106148, Proteintech, 1:1000); Pan-ADP-ribose (#MABE1016, Millipore, 1:1000); Poly/Mono-ADP-ribose (#83732, Lot #5, Cell Signaling Technology, 1:1000); Cyclin E2 (#4132S Lot #3, Cell Signaling Technology, 1:1000); HPF1 (#NBP1-93973 Lot #000004509, Novus, 1:1000); ARH3 (#sc-374162 Lot #G0919, Santa Cruz, 1:500); Tubulin (#T6557, Sigma-Aldrich, 1:5000); Poly-ADP-ribose (10H) (#ALX-804-220-R100, Enzo, 1:1000); FEN1 (#A300-256A, Lot #2, Fortis Life Sciences, 1:1000); DNA Ligase 1 (#A301-136A, Fortis Life Sciences, 1:1000); POT1 (#NB500-176, Novus, 1:1000); Myc (#MA1-21316, Thermo Fisher, 1:1000). |
| Validation | PAR westerns were validated by sensitivity to PARP inhibitor. TARG1, PARP1 antibody validated by knockout of corresponding protein. siRNA depletion of corresponding targets proteins was used in validation of other antibodies. where possible, previously validated antibodies were obtained from commercial vendors. Antibodies were validated by the suppliers companies for reactivity against the human proteins. |

# Eukaryotic cell lines

Policy information about cell lines and Sex and Gender in Research

| | |
|---|---|
| Cell line source(s) | U2OS (HTB-96) and IMR90 (CCL-96) cells were obtained from ATCC. Control and TARG1 KO U2OS and HeLa cells were generated and obtained from Ivan Ahel (Tromans-Coia C, Sanchi A, Moeller GK, Timinszky G, Lopes M, Ahel I, TARG1 protects against toxic DNA ADP-ribosylation, Nucleic Acids Research (2021)). Hek293E cell lines were obtained from Jayakrishna Nandakumar (Tesmer VM, Brenner KA, Nandakumar J. Human POT1 protects the telomeric ds-ss DNA junction by capping the 5' end of the chromosome. Science. (2023)). |
| Authentication | Cell lines validated by STR authentication at ATCC. |
| Mycoplasma contamination | Routinely tested negative using MycoAlert Kit (Lonza). Cells routinely treated my plasmocin prophylactic reagent. Incubators cleaned by high temp sanitization quarterly. |
| Commonly misidentified lines (See ICLAC register) | HeLa, HEK293E |

# Plants

| | |
|---|---|
| Seed stocks | N/A |
| Novel plant genotypes | N/A |
| Authentication | N/A |

