## [Peer Review File · Nature Structural & Molecular Biology]

Peer Review Information

Manuscript Title: Deregulated DNA ADP-Ribosylation impairs telomere replication

Corresponding author name(s): Roderick O'Sullivan

Reviewer Comments & Decisions:

Decision Letter, initial version:
--

Message: 12th Oct 2023

Dear Dr. O'Sullivan,

Thank you again for submitting your manuscript "Deregulated DNA ADP-Ribosylation impairs telomere replication". I apologise for the delay in responding, which resulted from the difficulty in obtaining suitable referee reports. Nevertheless, we now have comments (below) from the 3 reviewers who evaluated your paper. In light of these reports, we remain interested in your study and would like to see your response to the comments of the referees, in the form of a revised manuscript.

You will see that all experts appreciate the novel functional connection between DNA ADP ribosylation, via the PARP1/TARG axis, and telomere replication. However, all experts also raise concerns, at the technical and functional/mechanistic level, that need to be addressed in a revised manuscript. More specifically, the experts request further controls (we highlight R#1's request for excluding potential mitigating signal from protein ADP ribosylation), clarifications about certain data (R#2's question about the nature of the TARG1-KO being very important), and the addition of relevant primary data. As importantly, all referees raise pertinent functional questions, and associated experiments to address these questions, which we deem that, if answered, will significantly elevate the value of the manuscript. Amongst these requests, we note the common demand to adequately explain the functional roles of DNA ADP ribosylation, its frequency, its physiological significance in safeguarding the genome, its interplay with the DNA damage response, and its occurrence under different DNA damage sources.

Please be sure to address/respond to all concerns of the referees in full in a point-by-point response and highlight all changes in the revised manuscript text file.

We appreciate the requested revisions are extensive. We thus expect to see your revised manuscript within 6 months. If you cannot send it within this time, please let us know. We will be happy to consider your revision as long as nothing similar has been accepted for publication at NSMB or published elsewhere. Should your manuscript be substantially delayed without notifying us in advance and your article is eventually published, the received date would be that of the revised, not the original, version.

Reporting Summary:

When submitting the revised version of your manuscript, please pay close attention to our [href="https://www.nature.com/nature-portfolio/editorial-policies/image-integrity">Digital Image Integrity Guidelines](https://www.nature.com/nature-portfolio/editorial-policies/image-integrity). and to the following points below:

We require deposition of coordinates (and, in the case of crystal structures, structure

factors) into the Protein Data Bank with the designation of immediate release upon publication (HPUB). Electron microscopy-derived density maps and coordinate data must be deposited in EMDB and released upon publication. Deposition and immediate release of NMR chemical shift assignments are highly encouraged. Deposition of deep sequencing and microarray data is mandatory, and the datasets must be released prior to or upon publication. To avoid delays in publication, dataset accession numbers must be supplied with the final accepted manuscript and appropriate release dates must be indicated at the galley proof stage. Please find the complete NRG policies on data availability at <http://www.nature.com/authors/policies/availability.html>.

[Redacted]

Sincerely,

Dimitris Typas
Associate Editor
Nature Structural & Molecular Biology
ORCID: 0000-0002-8737-1319

Referee expertise:

Referee #1: PAR and PARP1 biology, in DNA damage context

Referee #2: Telomere biology and maintenance

Referee #3: PAR and PARP1 biology, structural biology

Reviewers' Comments:

Reviewer #1:

Remarks to the Author:

In this manuscript the authors describe a role of DNA ADP-ribosylation (DNA-ADPr) in telomere replication, which may have an impact on genome stability. DNA-ADPr has been previously described but where it is present in the genome, how it contributes to genome stability and which hydrolases regulate its turn-over, has remained unclear. With their work the authors claim that telomeres are the main subject to DNA-ADPr, and this is catalyzed by PARP1 and removed by TARG1. DNA-ADPr appears to be coupled to lagging telomere DNA strand synthesis, forming at single-stranded DNA present at unligated Okazaki fragments and on the 3' single-stranded telomere overhang. Persistent DNA-ADPr due to TARG1 deficiency leads to telomere shortening, which compromises telomere replication and integrity.

Overall, I think that the study provides useful insights that move the field of DNA ADP-ribosylation biology forward. With telomere replication the authors may have identified a central role for DNA-ADPr and thereby explain the importance of controlling DNA-ADP ribosylation turnover for sustained genome stability. Still, I have some concerns that I point out below. One major concern I have is that the authors firmly state that they detect DNA-associated ADPr in several of their assays. To me this is not always evident, especially in Figures 1+2. I understand that distinguishing protein from DNA ribosylation is a challenge and I appreciate the use of the elegant DarT-TRF1 fusion protein used in this study. Still, I wonder whether the data presented should not be described more carefully as chromatin-associated ADPr. The authors present strong data using state-of-the art biochemical tools to support their hypothesis. Maybe I am wrong and missed something, but currently, I would like to see a more careful interpretation, not excluding a role for protein-associated ADPr in several assays.

More specifically I have the following points to be addressed:

ED Fig. 1a.: The pronounced accumulation of ADPr foci in the TARG1-deficient cells colocalizing with telomeres is very intriguing. What happens in response to DNA damage, e.g. hydrogen peroxide, MMS or ionizing radiation? Does this result in additional foci outside of telomeres?

Fig. 1a: I find the RSE approach a useful biochemical tool. I just miss a control for the AATCCC-probe that detects telomeres. How would the data look like if random biotin-conjugated oligos are used?

Fig. 1: In general, I find the data showing PARP1 as the major catalyst of telomeric ADPr and TARG1 as the primary hydrolase responsible for removing ADPr solid and convincing. I am just wondering whether the authors can be sure that it is really ADP ribosylation of DNA? Does the RSE approach not pulldown chromatin and we may still be looking at ADPr of histones or PARP1 autoparylation? Especially PARylated proteins associated with ssDNA overhangs?

Fig. 2: Again very convincing mechanistic data to pinpoint an association of ADPr with with lagging strand telomere replication. Again my question: can the authors exclude a role for protein ADPr?

Figure 3: I find the DarT and DarG tools very elegant. Still, I do not understand the interpretation of Fig. 3c. The authors write "Importantly, western blot analysis confirmed that expression of DarT-TRF1 did not trigger widespread protein ADPr (Fig. 3c)". I see substantially increased ADPr in all of the conditions in which wt or mutant DarT or DarG is

expressed. Not as high as with H₂O₂ treatment, but still. I assume that it is autoparalyzed PARP1? Why is it activated? Is there still some unspecific damage induced when expressing these constructs? It is also unclear to me why there is less DarG protein when mutant DarT-TRF1 is expressed.

Figure 4: The results showing impairment of telomere integrity using the DarT tool are interesting. Still, I miss more data on the functional consequences of the TARG1 KO cells, independent of DarT or DarG. The clonogenic assay in Fig. 4e suggests slower growth of the TARG1 KO cells. Can the authors provide more data to confirm the consequences of the claimed genomic instability? According to depmap.org TARG1 is not essential in most cell lines. Is this not surprising if it has a central role in regulating telomere biology? What about deep sequencing of the TARG1 ko cells compared to non-targeting controls at equal passage number? Are there more mutations or specific signatures coming up?

Reviewer #2:

Remarks to the Author:

In "Deregulated DNA ADP-Ribosylation impairs telomere replication", Wondisford, Lee and colleagues investigate the effect of uncontrolled DNA ADP-ribosylation (ADPr) at telomeres.

First, they clearly detected ADPr specifically at human telomeric DNA and showed that such modification is promoted by PARP1 and promptly removed by PARG/TARG1. In fact, it can only be detected when TARG1 is deleted and/or PARG is inhibited. Using TARG1-deleted cells, the authors show that ADPr is induced in S-phase coupled with replication and increased by DNA damage and/or ssDNA accumulation due to ATR inhibition, unprocessed Okazaki fragments or Pot1 deletion/mutation. Consistently, it is detected on the newly replicated lagging strand telomeres, and mostly, but not all, on the overhang. Furthermore, they show that the inability of cells to remove ADPr causes several specific defects in telomere metabolism, as delay in replication at telomeres, lagging-strand telomere fragility, high telomere recombination and telomere shortening. To confirm that their findings are indeed due to DNA ADPr and not protein ADPr, the authors used a convincing combination of TRF1-DarT/DarG exogenous proteins, where DarT and DarG are of bacterial origin and specific for DNA.

While DNA PARylation was already shown, this is the first indication that physiological DNA PARylation occurs specific at telomeres and affects telomere metabolism, as far as this reviewer knows. Furthermore, with an elegant combination of genetics and chemical inhibitors they also define all the key players involved.

Overall, the manuscript increases our understanding of this recently discovered DNA modification. The experiments are well controlled, executed and presented. Data are robust and support the conclusions.

My main questions are:

1. Regarding the overhang, it is not completely clear if accumulation of ssDNA promotes ADPr and/or viceversa. For example, the increase in ssDNA in IMR90 cells after TARG1 deletion is not so evident in the blots shown in Fig. 1a, Fig. 2d and Fig. 2f. Furthermore, are TARG1KO cells a population or a clone? In case they are clones, it is very difficult to compare overhang signal between clones. Is the overhang increase

suppressed by their expression of TARG1?

2. Although TRF1-DarT increases RPA foci formation also in wild type cells (Fig.4b), it also seem to increase, rather than decrease, the replication rate in wt cells (Fig.4a). Is the increase significant? How could it be explained?

3. Is there any activation of DNA damage response when DNA ADPr is not removed in TARG1KO cells? While there is no Chk1 or RPA phosphorylation, could it be that there is accumulation of γ -H2AX or 53BP1 foci at telomeres, at least in S-phase? Also, I noticed that TARG1KO cells have less γ -H2AX than wild type cells (ED Fig.4b), how could this correlate with more telomere fragility?

MINOR POINTS:

1. ED Fig. 1c: PARG seems to be involved in the removal of ADPr too. Does PARGi also cause telomere defects?

2. Pag. 5, line 24: why the sensitivity to exo1 should indicate lagging-strand overhang and not just the overhang? Was the blot performed on separate leading versus lagging DNA? Pot1 blocks Apollo and promotes CST-dependent fill-in at both lagging and leading-end telomeres (DOI: 10.1016/j.cell.2012.05.026)

3. Fig. 2f: IMR90: why the overhang runs so differently from total telomere?

4. Fig. 4e-f: Expression of DarG suppress the decrease survival in TARG1KO cells. Does PARPi do the same?

COMMENTS TO FIGURES/TEXT:

1. High levels of ADPr are clearly a problem for the cells, and they are never observed in the presence of functional TARG1. Could the authors comment on why ADPr is occurring, if with a reason/purpose or just as a deleterious side-product of replication?

2. Furthermore, DNA ADPr is not only present on ssDNA, but also on dsDNA. Could the authors comment on this?

3. The paper is well written, but it is a very complex story; it would benefit to add a final model.

4. Indicate the figure the cells used in each panel. This would make it clearer to understand when the experiment is performed only in TARG1KO cells.

5. While most of the data are clear, some would benefit from further quantifications. For example, Fig. 2b, Fig. 2f (the increase of overhang is not always so visible) or ED Fig. 1c (the additive affect is not so clear).

6. Fig. 1a: Adding the DNA at the schematics would add clarity to the explanation of the protocol.

7. Fig 2a: It would be good to show the original blots from which the quantification was done.

8. Fig. 2f: change E, M, and L with the exact time points, possibly as PDs.

9. Fig. 2f: The more extreme telomere shortening observed in U2OS could be due to DNA ADPr affecting more strongly ALT?

10. ED Fig. 2c does not seem to be called in the text.

11. Fig 4a: the media (or average) of the distributions are not so clear, would it be possible to add the number to the graph?

12. While the TRF1-DarT/DarG system is very elegant, it clearly pushes the phenotype. Would it be possible to move the telomere phenotypes of TARG1KO cells from ED (ED Fig. 4c-f) to main figures?

Reviewer #3:

Remarks to the Author:

O'Sullivan and colleagues provide evidence for ADP-ribose modification of telomeric DNA. Their proposed model is that PARP1 generates this modification on single-stranded regions of lagging strand telomeric DNA during S phase, and that the enzyme TARG1 reverses the modification to avoid the problems it seems to cause with telomere replication. Most of the analysis focused on TARG1 knockout cell lines that showed the clearest evidence for the ADP-ribose modification in the isolated telomeric DNA. The nature of the ADP-ribose-DNA modification is not clearly determined, but the use of the enzyme DarT that specifically modifies thymidine bases in single-stranded DNA suggested that this might be the type of ADP-ribose modification being produced by PARP1 (TARG1 appears to reverse this thymidine modification). There are several experiments that suggest that deficient lagging strand processing of Okazaki fragments generates the ADP-ribose-DNA modification. The experiments are of high quality and well controlled, and the conclusions of the study match the results for the most part. The study will add to the growing appreciation that ADP-ribose modification of DNA is more prevalent than previously appreciated.

The presented model for PARP1/TARG1 regulation of ADP-ribose-DNA is compelling. However, the manuscript lacks some description of why this modification exists, and how frequently does it occur. Is it a mistake that happens simply because PARP1 is so abundant and happens to modify single-stranded DNA when there are elevated levels? If so, is this a rare event? The Discussion seems to indicate that this modification is good for the genome:

"Furthermore, our ability to manipulate DNA-ADPr catalysis specifically at telomeres with DarT/DarG enabled us to uncover a direct role for this modification in safeguarding telomeres"

However, this statement is not so evident to me and would need more explanation. How does the modification safeguard the genome?

Another Discussion comment is also confusing:

"the failure to efficiently remove DNA-linked ADPr disrupts telomere replication by uncoupling telomere DNA strand synthesis"

I am not sure what uncoupling refers to - is it meant to say that the effect is only on the lagging strand?

"This was despite the evidence that acute DarT-TRF1 expression only provoked modest DDR signaling with increased RPA2 and H2AX phosphorylation (Extended Data Fig. 4b)." The referenced data requires some interpretation to arrive at the conclusion in this sentence. There are changes in the gamma-H2AX levels that were not easy for me to understand.

Some of the dot blots were quantified, while others were not, even if there were some comments made about relative levels. For thoroughness, all of the dot blots should be quantified.

Consider including the primary dot blot data used to generate the plot in Fig. 2a.

"expression of POT1-F62A, a POT1 ssDNA binding whose expression hyperextends" ssDNA binding mutant

Impaired telomere replication section: Fig. 2e is referenced a couple of times when I believe that 2f was intended.

Figure 3c. What is the reason for the lower levels of DarG/FLAG in the last four lanes of the gel? I could not figure out why this would be expected.

I am more accustomed to seeing the abbreviation ADPr used to mean ADP-ribose, whereas this manuscript defines it to mean ADP-ribosylation. That gets a little confusing when talking about the modification versus the activity, and it seems to be used in both senses in the manuscript.

Author Rebuttal to Initial comments

Reviewer #1: PAR and PARP1 biology, in DNA damage context

Remarks to the Author:

In this manuscript the authors describe a role of DNA ADP-ribosylation (DNA-ADPr) in telomere replication, which may have an impact on genome stability. DNA-ADPr has been previously described but where it is present in the genome, how it contributes to genome stability and which hydrolases regulate its turn-over, has remained unclear. With their work the authors claim that telomeres are the main subject to DNA-ADPr, and this is catalyzed by PARP1 and removed by TARG1. DNA-ADPr appears to be coupled to lagging telomere DNA strand synthesis, forming at single-stranded DNA present at unligated Okazaki fragments and on the 3' single-stranded telomere overhang. Persistent DNA-ADPr due to TARG1 deficiency leads to telomere shortening, which compromises telomere replication and integrity.

Overall, I think that the study provides useful insights that move the field of DNA ADP-ribosylation biology forward. With telomere replication the authors may have identified a central role for DNA-ADPr and thereby explain the importance of controlling DNA-ADP ribosylation turnover for sustained genome stability. Still, I have some concerns that I point out below. One major concern I have is that the authors firmly state that they detect DNA-associated ADPr in several of their assays. To me this is not always evident, especially in Figures 1+2. I understand that distinguishing protein from DNA ribosylation is a challenge and I appreciate the use of the elegant DarT-TRF1 fusion protein used in this study. Still, I wonder whether the data presented should not be described more carefully as chromatin-associated ADPr. The authors present strong data using state-of-the-art biochemical tools to support their hypothesis. Maybe I am wrong and missed something, but currently, I would like to see a more careful interpretation, not excluding a role for protein-associated ADPr in several assays.

We thank the reviewer for their on-point comments. We addressed those as thoroughly as we could. Our responses to the issues raised are below, and we hope they are satisfactory. The reviewer's major question is whether we are truly detecting nucleic acid linked ADPr. Below we provide further experimental evidence and commentary that support this. We hope this clarifies the issue. The other queries are addressed. Thank you.

More specifically I have the following points to be addressed:

ED Fig. 1a.: The pronounced accumulation of ADPr foci in the TARG1-deficient cells colocalizing with telomeres is very intriguing. What happens in response to DNA damage, e.g. hydrogen peroxide, MMS or ionizing radiation? Does this result in additional foci outside of telomeres?

Response: We treated TARG1 KO U2OS with hydrogen peroxide.

This resulted in global or pan-nuclear ADPr signals. This likely reflects the massive poly-ADP-ribosylation induction after systemic DNA damage. Examples of what is observed are shown here. Our interpretation is that the focal enrichment of telomere ADPr in ED. Fig1a suggests that telomeres in TARG1 KO cells are subject to persistent and chronic, low-level replicative stress. We believe this emanates from difficulty in replicating through long arrays of repetitive sequences and processing of the overhang.

Fig. 1a: I find the RSE approach a useful biochemical tool. I just miss a control for the AATCCC-probe that detects telomeres. How would the data look like if random biotin-conjugated oligos are used?

Response: The RSE methodology has previously been employed and validated in a publication from our collaborator Dr. Opresko (Parikh et al., 2015). A scrambled control oligo was used, which did not pulldown telomeric DNA. In optimizing the methodology, we also employed this scrambled oligo in our experiments and again did not enrich for telomeric DNA. The result is shown in the adjacent figure. As our colleague has published this, we will not include this in the revised paper, but the paper is cited.

Ref; Parikh et al. Nature Communications (2015)

Fig. 1: In general, I find the data showing PARP1 as the major catalyst of telomeric ADPr and TARG1 as the primary hydrolase responsible for removing ADPr solid and convincing. I am just wondering whether the authors can be sure that it is really ADP ribosylation of DNA? Does the RSE approach not pulldown chromatin and we may still be looking at ADPr of histones or PARP1 autoparylation? Especially PARylated proteins associated with ssDNA overhangs?

Response: We have used extensive control in establishing and validating the procedure. These show that the ADPr signals observed are derived from DNA, not chromatin proteins or histones.

- Cell pellets were treated with proteinase K and RNaseA during lysis.
- DNA was purified by phenol/chloroform/isoamyl separation and Ethanol precipitation.
- Following biotin pulldown, telomere DNA was treated with Proteinase K and RNaseA and column purified.
- As shown in **Fig1a** and **ED Fig1b**, the ADPr signals are sensitive to DNaseI and benzonase. In the adjacent figure, we show he ADPr signals are insensitive to proteinase K.
- In **ED Fig1c**, we show that depletion of HPF1, the essential cofactor for chromatin PARYlation (Suskiewicz et al., Bilokapic et al. 2020), did not affect the ADPr signals.
- In Fig 3, we show that DNA-ADPr with DarT-TRF1 does not stimulate histone ADPr (i.e., no bands at ~15 kDa).

Refs; Suskiewicz et al., Nature (2020); Bilokapic et al., Nature (2020)

Fig. 2: Again very convincing mechanistic data to pinpoint an association of ADPr with with lagging strand telomere replication. Again, my question: can the authors exclude a role for protein ADPr?

Response: We added the density readings over the graph in Fig.2a.

Cellular macromolecules have distinct densities that enable differential fractionation through CsCl gradient centrifugation. The accompanying image displays the densities at which protein and DNA fractionate. The CsCl density ranges of fractions used in Fig.2a are between 1.7-1.8 g/ml: RNA fractionates at the bottom of the gradient, protein at the top – outside the density range collected for the DNA samples so they are not fractionated. As indicated in the prior response, the DNA was also treated with Proteinase K, RNaseA, and phenol-chloroform before and after centrifugation. We are confident that the ADP-ribosylation signals come from DNA, not Protein or RNA.

Figure 3: I find the DarT and DarG tools very elegant. Still, I do not understand the interpretation of Fig. 3c. The authors write “Importantly, western blot analysis confirmed that expression of DarT-TRF1 did not trigger widespread protein ADPr (Fig. 3c)”. I see substantially increased ADPr in all of the conditions in which wt or mutant DarT or DarG is expressed. Not as high as with H₂O₂ treatment, but still. I assume that it is autophosphorylated PARP1? Why is it activated? Is there still some unspecific damage induced when expressing these constructs? It is also unclear to me why there is less DarG protein when mutant DarT-TRF1 is expressed.

Responses:

- 1) We believe that this may be auto-PARylated PARP1 that is induced by transfection – as has been shown in prior studies. The 4 lanes that do not show this are proteins from cells that were not transfected with DNA or exposed to transfection reagents. Please note that PARPi/PARGi were NOT added to the lysis buffer as we did in other experiments. This was deliberate since the intended purpose was to assess potential widespread PARP1 activation on chromatin and other protein targets directly or indirectly following DarT/DarG expression. Thus, adding PARPi/PARGi to the lysis buffers here would have been potentially misleading. We have added this information to the methods. As DarT localization and activity are unaffected by PARPi (ED. Fig.3) we can rule out any impact of PARP1 auto-PARylation on our results as they pertain to telomeric DNA and not protein as indicated in the prior response. We reworded the statement to reflect the absence of PAR smears and the absence of chromatin ADPr (i.e., on histones).
- 2) We also noted the effect on DarG when mutant DarT-TRF1 is expressed. It is intriguing. DarT and DarG physically interact as an obligate heterodimeric complex. The catalytic ADP-ribosyltransferase (ART) loop which is mutated (E183A) disrupts contacts with the DarG TBD domain. So, we hypothesize that mutant DarT-E183A affects not only the ART catalytic activity but also the binding and stability of DarG. Future studies requiring a detailed structural and biochemical analysis of DarT/DarG interactions will assess this. We respectfully assert that this is beyond this study's scope but does not impact our results or conclusions.

Refs; Jankevicius et al., Mol.Cell (2016), Schuller et al. Nature (2021), Deep et al. Structure (2023)

Figure 4: The results showing impairment of telomere integrity using the DarT tool are interesting. Still, I miss more data on the functional consequences of the TARG1 KO cells, independent of DarT or DarG. The clonogenic assay in Fig. 4e suggests slower growth of the TARG1 KO cells. Can the authors provide more data to confirm the consequences of the claimed genomic instability? According to depmap.org TARG1 is not essential in most cell lines. Is this not surprising if it has a central role in regulating telomere biology? What about deep sequencing of the TARG1 ko cells compared to non-targeting controls at equal passage number? Are there more mutations or specific signatures coming up?

Responses:

- 1) In **ED Fig.2g,h** we provide new data showing a modest 53BP1 accumulation at telomeres in S-phase and a greater frequency of micronuclei containing RPA and telomere fragments in TARG1 KO U2OS cells. We posit that although 53BP1 can accumulate at telomeres, it does not at levels that reflect the presence of an overwhelming telomere DDR response. However, the clear increases in micronuclei (**ED. Fig2h**), including those with RPA2 and telomeric DNA, suggest that the cells acquire significant genome instability, likely later in mitosis.
 - 2) We are not surprised that TARG1 KO is non-essential in most cell lines yet has an important role in controlling telomere replication. PARP1 and other PARPs, as well as ADPr-hydrolases like PARG, ARH3, MACROD1 are also non-essential per depmap, have important functions in telomere and genome stability and are of high biomedical & therapeutic value. Recent studies from the Caldecott and Ahel labs have provided clear demonstrations of synergism and redundancy among the ADPr hydrolases – where the loss of one enables another to step in, albeit often with less efficiency thus rendering cells susceptible to further genotoxic insults.
- Refs; Hanzlikova et al., Nat Comms (2020), Prokhorova et al., Molecular Cell (2022), Gros Lambert et al., Cell Reports 2023.
- 3) The reviewer's suggestion that TARG1 KO cells could bear mutational signatures is intriguing. We believe conducting these analyses would likely prove highly complex and require extensive bioinformatic investment. We believe that this is beyond the scope of this initial manuscript. But we agree this could be a nice future study and have plans to pursue this.

Referee #2: Telomere biology and maintenance

Remarks to the Author:

In “Deregulated DNA ADP-Ribosylation impairs telomere replication”, Wondisford, Lee and colleagues investigate the effect of uncontrolled DNA ADP-ribosylation (ADPr) at telomeres.

First, they clearly detected ADPr specifically at human telomeric DNA and showed that such modification is promoted by PARP1 and promptly removed by PARG/TARG1. In fact, it can only be detected when TARG1 is deleted and/or PARG is inhibited. Using TARG1-deleted cells, the authors show that ADPr is induced in S-phase coupled with replication and increased by DNA damage and/or ssDNA accumulation due to ATR inhibition, unprocessed Okazaki fragments or Pot1 deletion/mutation. Consistently, it is detected on the newly replicated lagging strand telomeres, and mostly, but not all, on the overhang. Furthermore, they show that the inability of cells to remove ADPr causes several specific defects in telomere metabolism, as delay in replication at telomeres, lagging-strand telomere fragility, high telomere recombination and telomere shortening. To confirm that their findings are indeed due to DNA ADPr and not protein ADPr, the authors used a convincing combination of TRF1-DarT/DarG exogenous proteins, where DarT and DarG are of bacterial origin and specific for DNA.

While DNA PARylation was already shown, this is the first indication that physiological DNA PARylation occurs specific at telomeres and affects telomere metabolism, as far as this reviewer knows. Furthermore, with an elegant combination of genetics and chemical inhibitors they also define all the key players involved.

Overall, the manuscript increases our understanding of this recently discovered DNA modification. The experiments are well-controlled, executed and presented. Data are robust and support the conclusions.

We thank the reviewer for their positive comments and critiques. We sought to address these to the best of our ability in the revised manuscript. We addressed several critiques experimentally, as well as those requiring clarification and explanation. These include:

- Western analyses of DDR signaling in TARG1 KO cells
- 53BP1 foci analysis in TARG1 KO cells
- Micronuclei analysis in TARG1 KO cells

Furthermore, prompted by the reviewers' astute suggestion, we can now show that the increased ss overhang detected in TARG1 KO cells can be restored to normal upon expressing WT-TARG1. This provides additional evidence for TARG1 (and DNA-ADPr) having a direct role in telomere overhang regulation. This is shown in **ED Fig.2**.

Lastly, per the reviewer's advice, the revised manuscript now contains an additional figure containing the illustrated model detailing our summary/working hypotheses (**ED. Fig.5**). We hope it is satisfactory but welcome any additional advice.

My main questions are:

1. Regarding the overhang, it is not completely clear if the accumulation of ssDNA promotes ADPr and/or vice versa. For example, the increase in ssDNA in IMR90 cells after TARG1 deletion is not so evident in the blots shown in Fig. 1a, Fig. 2d and Fig. 2f. Furthermore, are TARG1KO cells a population or a clone? In case they are clones, it is very difficult to compare overhang signal between clones. Is the overhang increase suppressed by the re-expression of TARG1?

Responses:

- 1) The outcome of FEN1i and POT1-F62A experiments indicates that DNA-ADPr and ssDNA abundance are closely correlated. This is consistent with the structural work from DarT, the bacterial precursor of PARP, showing a selective sensing and modifying activity on ssDNA (Schuller et al., Deep et al). Thus, we propose that it is the availability of ssDNA that stimulates DNA-ADPr.
- 2) We agree that the increase in ssDNA is not so readily obvious in IMR90-E6E7, particularly in Fig.1a. We hope the quantification now shown helps to clarify this. This shows a 25-30% increase in ssDNA in telomere DNA from TARG1 KO cells. The gel image and quantification of the native/denatured ratio in Fig 2d show a robust increase in the overhang in the IMR90-E6E7.
- 3) For the reasons stated by the reviewer, all overhang and telomere length-based assays are conducted in cell populations, not clones.
- 4) In ED Fig2d, we show that the re-expression of TARG1 restored the normal ratio of native to denatured DNA signal. The catalytic mutant, K84A, did not. This indicates that TARG1 deficiency does directly impact overhang regulation.

Refs; Schuller et al. Nature (2021), Deep et al. Structure (2023)

2. Although TRF1-DarT increases RPA foci formation also in wild-type cells (Fig.4b), it also seems to increase, rather than decrease, the replication rate in wt cells (Fig.4a). Is the increase significant? How could it be explained?

Response:

We are certain that the perceived increase in replication rate in wt-cells is not the case. When we assess the SMAT results in the CTRL cell conditions, we can see that both parental and wt-DarT-TRF1 conditions are highly consistent. However, in 1 of the 3 experiments where mut-DarT-TRF1 was expressed in CTRL U2OS cells, we found that the average EdU tract length was notably shorter. This was due to a greater proportion of shorter fibers in that set of ~150 fibers. This single series could be considered an outlier. However, each set (wt/mut in CTRL/TARG1 KO) of experiments was performed concurrently, and all other samples were highly consistent across independent sets of experiments, especially the in TARG1 KO cells. Thus, the data are represented as such.

Based on the comparison with the parental CTRL line we are sure that telomere replication in wild-type (CTRL) cells is not increased by WT-DarT-TRF1 but rather the decrease in CTRL with EA-DarT-TRF1 is due to this experimental 'outlier'. However, this does not impact the key findings that WT-DarT-TRF1 reduced replication rate in TARG1 KO cells, confirmed in the normalized data shown in Fig 4a and in the adjacent figure when individual experiments are averaged.

3. Is there any activation of DNA damage response when DNA ADPr is not removed in TARG1KO cells? While there is no Chk1 or RPA phosphorylation, could it be that there is accumulation of γ -H2AX or 53BP1 foci at telomeres, at least in S-phase? Also, I noticed that TARG1KO cells have less γ -H2AX than wild type cells (ED Fig.4b), how could this correlate with more telomere fragility?

Responses:

1) Our data and that of our collaborator show that TARG1 deletion alone does not induce a strong DDR. Our western and cell cycle analyses shown in **ED Fig.4b,c** confirm this, even though the cells proliferate a little slower (**Fig.4f**). Concerning γ H2AX levels in TARG1 KO cells, we believe the underloading of prior blots would have misled the reviewer. Thus, we repeated these blots. New, equally loaded blots are now presented in **ED Fig.4b**. As full-blown ATR activation requires an exposed 5' end at the double-stranded–single-stranded junction (MacDougall et al. 2007), accumulating single-stranded DNA on the lagging strand may not activate the DDR in TARG1 or DarT-TRF1 expressing cells. The experiment with the 'POTHOLE' R83E and F62A POT1 mutants in **ED Fig3** shows that ssDNA and DNA-ADPr accumulation are highly correlated.

Please note that our recent collaborative paper showed that the addition of PARG inhibitor to TARG1 KO cells stimulates a strong ATR kinase-dependent DDR. This paper also showed that TARG1 KO cells exhibit sensitivity to Etoposide and ATRi (Gros Lambert et al., 2023). Thus, it appears the TARG1 KO cells are primed for checkpoint activation due to replicative stress, but that TARG1 deficiency alone is not enough to trigger robust DDR activation.

2) We conducted IF-FISH of 53BP1 colocalization with telomeres in asynchronous and S-phase synchronized CTRL and TARG1 KO U2OS cells. While no distinction was observed in asynchronous cells, there was a ~2-fold increase in 53BP1-telomere colocalizations in TARG1 KO U2OS cells. Considered in conjunction with the cell cycle and western blot analysis, we assert that this modest increase does not reflect a robust telomere DDR or telomere deprotection due to TARG1 deficiency. These new data are shown in **ED Fig.2i**. the increased ssDNA and elevated binding RPA at telomeres are all consistent with a mild replicative stress that manifests as telomere fragility, without checkpoint activation.

Refs; Gros Lambert et al., Cell Reports (2023), MacDougall et al. Genes and Dev (2007)

MINOR POINTS:

1. ED Fig. 1c: PARG seems to be involved in the removal of ADPr too. Does PARGi also cause telomere defects?

Response: Prior published studies from our group have shown that PARGi can suppress the ALT mechanism (Hoang et al., 2020). Subsequent published studies from the Ahel lab have shown that accumulating toxic protein ADPr due to combined ARH3 and PARGi enhances the inhibition of ALT (Prokhorova et al., 2021). The Opresko lab has shown that PARGi can prevent the repair of 8-oxo guanine due to misprocessing of replication intermediates (Fouquerel et al., 2018). Whether PARGi affects telomere protection is unknown even though PARP1-Lig3 is implicated in telomere fusion by alt-NHEJ (Sfeir et al., 2012), suggesting that PARGi which would maintain PARP1-mediated ADPr could have some inhibitory effect on telomere fusion by alt-NHEJ. The latter remains to be determined.

Refs; Hoang et al., Nature Structure & Molecular Biology (2020), Prokhorova et al., Mol. Cell (2021), Fouquerel et al., (2019), Sfeir et al., Science (2012)

2. Pag. 5, line 24: why the sensitivity to exo1 should indicate lagging-strand overhang and not just the overhang? Was the blot performed on separate leading versus lagging DNA? Pot1 blocks Apollo and promotes CST-dependent fill-in at both lagging and leading-end telomeres (DOI: 10.1016/j.cell.2012.05.026)

Response: The assays were not performed on separated DNA strands. This wording was my error that has been corrected. Now only the overhang is referred to. Apologies.

3. Fig. 2f: IMR90: why the overhang runs so differently from total telomere?

Response: We do not know, but this is what has been repeatedly observed. Below, we show unprocessed tiff images from the typhoon scanner so the reviewer can assess the images (IMR90 on left in both). Similar results were obtained with different DNA preps. During revision, we repeated the PFGE on new DNA preps, and obtained the same outcome.

4. Fig. 4e-f: Expression of DarG suppress the decreased survival in TARG1KO cells. Does PARPi do the same?

Response: TARG1 KO U2OS cells exhibit greater sensitivity to PARPi (Olaparib) than controls. This was shown in our recent study (Gros Lambert et al., 2023). The data are shown over for your convenience.

Ref; Gros Lambert et al., Cell Reports (2023),

COMMENTS TO FIGURES/TEXT:

1. High levels of ADPr are clearly a problem for the cells, and they are never observed in the presence of functional TARG1. Could the authors comment on why ADPr is occurring, if with a reason/purpose or just as a deleterious side-product of replication?

Responses:

- 1) That ADPr is not observed in the presence of functional TARG1 is consistent with all ADPr species (MAR and PAR) existing very transiently, being highly regulated to maintain cellular NAD⁺ levels.
- 2) Our study supports the growing appreciation that ADPr, and now DNA-ADPr, is a normal feature of DNA and telomere replication, specifically in regulating the temporal maturation of Okazaki fragments and lagging DNA strands. However, when unhydrolyzed, persistent DNA-ADPr represents an obstacle potentially leading to uncoupling of DNA replication dynamics and genome instability, including at telomeres. We hope the edited text and model in **ED Fig5** help to clarify this.

2. Furthermore, DNA ADPr is not only present on ssDNA, but also on dsDNA. Could the authors comment on this?

Response: We are unsure what data the reviewer is referring to. Possibly the dsDNA blot in **Fig.1a**? If so, this dsDNA blot shows that the RSE method efficiently captures telomeric dsDNA. If we are mistaken, then our data strongly point to ssDNA being the major substrate for PARP1/TARG1-regulated DNA-ADPr. The data derived from the experiments with FEN1i and POT1 mutants support this. Furthermore, the structural, genetic, and biochemical characterization of the DarT/DarG toxin-antitoxin system conducted to date has thoroughly and conclusively defined thymidine bases in ssDNA as the precise substrate of DarT-dependent ART activity. That DarG removes all ADPr (i.e., baseline TARG1 KO ADPr & DarT-TRF1 induced (see **Fig.3e – columns 3, 4 & 6**)) supports that ssDNA is the substrate to which ADPr is covalently linked.

Refs; Jankevicius et al., Mol.Cell (2016), Schuller et al. Nature (2021), Deep et al. Structure (2023)

3. The paper is well written, but it is a very complex story; it would benefit to add a final model.

Response: Yes. We added a model figure to **ED Fig.5**. We hope this is now satisfactory.

4. Indicate the figure the cells used in each panel. This would make it clearer to understand when the experiment is performed only in TARG1KO cells.

Response: No experiments are conducted only in TARG1 KO cells. But to assist we indicate the cell lines used in the figures where necessary and in the legends. Hope this is ok.

5. While most of the data are clear, some would benefit from further quantifications. For example, Fig. 2b, Fig. 2f (the increase of overhang is not always so visible) or ED Fig. 1c (the additive affect is not so clear).

Response: Graphs of quantified data now accompany all dot blots in the paper. Thank you.

6. Fig. 1a: Adding the DNA at the schematics would add clarity to the explanation of the protocol.

Response: We have added DNA to the schematic. Hope it is clear. Thank you.

7. Fig 2a: It would be good to show the original blots from which the quantification was done.

Response: The original blots are added to **Fig.2a**. Thank you.

8. Fig. 2f: change E, M, and L with the exact time points, possibly as PDs.

Response: This has been done. The PDs following delivery of sgRNAs are indicated.

9. Fig. 2f: The more extreme telomere shortening observed in U2OS could be due to DNA ADPr affecting more strongly ALT?

Response: Indeed. This is highly likely and supported by our assessment of EdU incorporation at telomeres shown in **ED Fig. 4a**. We are generating additional data for a separate follow-up comprehensive manuscript about the impact of TARG1 deficiency and DNA-ADPr specifically on ALT. Respectfully, presenting all those data and results is far beyond the scope of this manuscript but can be appropriately addressed in that subsequent publication.

10. ED Fig. 2c does not seem to be called in the text.

Response: This has been amended. Thank you.

11. Fig 4a: the media (or average) of the distributions are not so clear, would it be possible to add the number to the graph?

Response: We amended the figure to help clarify and added the numerical values to the graphs. Thank you.

12. While the TRF1-DarT/DarG system is very elegant, it clearly pushes the phenotype. Would it be possible to move the telomere phenotypes of TARG1KO cells from ED (ED Fig. 4c-f) to main figures?

Response: Our thoughts are that this figure pertains to DarT-TRF1, so we chose only to include data from experiments performed with this system. We integrated the data as the reviewer suggested but found that moving these data and figures makes the main figure far too dense. Also, the data with *mut-DarT-TRF1* is almost identical to that obtained in the parental TARG1 KO U2OS cells. We hope the reviewer will see and support our reasoning for leaving the data as presented in the original manuscript.

Referee #3: PAR and PARP1 biology, structural biology

Remarks to the Author:

O'Sullivan and colleagues provide evidence for ADP-ribose modification of telomeric DNA. Their proposed model is that PARP1 generates this modification on single-stranded regions of lagging strand telomeric DNA during S phase, and that the enzyme TARG1 reverses the modification to avoid the problems it seems to cause with telomere replication. Most of the analysis focused on TARG1 knockout cell lines that showed the clearest evidence for the ADP-ribose modification in the isolated telomeric DNA. The nature of the ADP-ribose-DNA modification is not clearly determined, but the use of the enzyme DarT that specifically modifies thymidine bases in single-stranded DNA suggested that this might be the type of ADP-ribose modification being produced by PARP1 (TARG1 appears to reverse this thymidine modification). There are several experiments that suggest that deficient lagging strand processing of Okazaki fragments generates the ADP-ribose-DNA modification. The experiments are of high quality and well controlled, and the conclusions of the study match the results for the most part. The study will add to the growing appreciation that ADP-ribose modification of DNA is more prevalent than previously appreciated.

We greatly appreciate the reviewers for their nice comments regarding this study. Thank you. Our responses to the very on-point comments are below. Overall, where possible, we addressed these by providing new data and clarifying prior data. Several reviewer comments relate to the discussion and larger "meta" questions that will occupy the field for future years. We addressed these but wanted to be cautious not to overstep or overstate. We hope that the reviewer agrees.

The presented model for PARP1/TARG1 regulation of ADP-ribose-DNA is compelling. However, the manuscript lacks some description of why this modification exists, and how frequently does it occur. Is it a mistake that happens simply because PARP1 is so abundant and happens to modify single-stranded DNA when there are elevated levels? If so, is this a rare event? The Discussion seems to indicate that this modification is good for the genome:

"Furthermore, our ability to manipulate DNA-ADPr catalysis specifically at telomeres with DarT/DarG enabled us to uncover a direct role for this modification in safeguarding telomeres"

However, this statement is not so evident to me and would need more explanation. How does the modification safeguard the genome?

Another Discussion comment is also confusing:

"the failure to efficiently remove DNA-linked ADPr disrupts telomere replication by uncoupling telomere DNA strand synthesis"

I am not sure what uncoupling refers to - is it meant to say that the effect is only on the lagging strand?

Response: We addressed the reviewer's comments in the revised manuscript and discussion. Furthermore, prompted by reviewer 2, the revised manuscript includes an illustrated model (ED Fig.5) depicting our current working hypotheses. We believe the reviewer's comments are addressed therein and hope these are satisfactory.

"This was despite the evidence that acute DarT-TRF1 expression only provoked modest DDR signaling with increased RPA2 and H2AX phosphorylation (Extended Data Fig. 4b)."

The referenced data requires some interpretation to arrive at the conclusion in this sentence. There are changes in the gamma-H2AX levels that were not easy for me to understand.

Response: Apologies for this confusion. We revisited this by repeating the western blot analysis. The data

show a modest increase in phospho-RPA2 and H2AX in TARG1 KO cells that is enhanced following DarT-TRF1. However, we do not observe the induction of phospho-CHK1-317. This indicates some degree of genomic stress, but it is insufficient to induce ATR checkpoint signaling in TARG1 KO cells, and following DarT-TRF1 expression. These data are consistent with prior data from the Cimprich lab documenting that ssDNA is not sufficient to induce ATR-dependent DDR signaling (MacDougall et al., 2007)

Ref, MacDougall et al. Genes and Dev (2007)

Some of the dot blots were quantified, while others were not, even if there were some comments made about relative levels. For thoroughness, all of the dot blots should be quantified.

Response: The quantifications of all dot blots are presented in the revised manuscript. Thank you.

Consider including the primary dot blot data used to generate the plot in Fig. 2a.

Response: Yes. These have been added to **Fig.2a**. Thank you.

"expression of POT1-F62A, a POT1 ssDNA binding whose expression hyperextends"
ssDNA binding mutant

Response: This has been corrected. Thank you.

Impaired telomere replication section: Fig. 2e is referenced a couple of times when I believe that 2f was intended.

Response: This has been corrected. Thank you.

Figure 3c. What is the reason for the lower levels of DarG/FLAG in the last four lanes of the gel? I could not figure out why this would be expected.

Response: We also noted the effect on DarG when mutant DarT-TRF1 is expressed. It is intriguing. DarT and DarG physically interact as an obligate heterodimeric complex as previously implied (Schuller et al., 2021). Recent structural studies have confirmed that the mutated ART loop (E183A) disrupts contacts with the DarG TBD domain (Deep et al., 2023). So, we hypothesize that mutant DarT-E183A affects not only the ART catalytic activity but also the binding and stability of DarG. Future studies requiring a detailed structural and biochemical analysis of DarT and DarG interactions will assess this. We respectfully assert that this is beyond this study's scope and does not impact our results or conclusions.

Refs; Schuller et al. Nature (2021), Deep et al. Structure (2023)

I am more accustomed to seeing the abbreviation ADPr used to mean ADP-ribose, whereas this manuscript defines it to mean ADP-ribosylation. That gets a little confusing when talking about the modification versus the activity, and it seems to be used in both senses in the manuscript.

Response: We applied the definitions and abbreviations of ADPr that were used in recent publications from the Ahel lab. ADPr refers to ADP-ribosylation. ADP-ribose and its various forms (mono, poly, pan) are explicitly indicated in the text. We hope this is reasonable and satisfactory.

Decision Letter, first revision:

Message: Our ref: NSMB-A48187A

26th Jan 2024

Dear Dr. O'Sullivan,

Thank you for submitting your revised manuscript "Deregulated DNA ADP-Ribosylation impairs telomere replication" (NSMB-A48187A). It has now been seen by the original referees and their comments are below. The reviewers find that the paper has improved in revision, and therefore we are happy to accept it in principle in Nature Structural & Molecular Biology, pending minor revisions to satisfy the referees' final requests and to comply with our editorial and formatting guidelines.

We are now performing detailed checks on your paper and will send you a checklist detailing our editorial and formatting requirements in about two weeks. Please do not upload the final materials and make any revisions until you receive this additional information from us.

Sincerely,

Dimitris Typas
Associate Editor
Nature Structural & Molecular Biology
ORCID: 0000-0002-8737-1319

Reviewer #1 (Remarks to the Author):

I am satisfied with the responses to my comments. Thank you for this exciting work!

Reviewer #2 (Remarks to the Author):

I would like to thank the authors for considering my comments. The revised manuscript addresses all my concerns. The further analysis of DDR in TARG1 KO cells is convincing, the quantifications are clear, and the model is compelling.

I have few suggestions:

1. Extended Data Fig.1a: is + PARGi? Also PARG is not really introduced in the text.
2. There are still some points in the text where POT1 is indicated as regulating specifically the "lagging" strand replication (Page 5, line 16, 19), when POT1 regulates the overhang metabolism after replication on both strands.
3. POT1-F62A does not seem to be completely "unable to" suppress the DNA-ADPr formation (Page 6, line 5 and Extended data Fig.2h). Which is the p-value compared to

POT1 KO?

4. Extended data Fig.2h: shouldn't the p-value be shown compared to TARG1 siRNA instead than NT siRNA?

5. Figure 3a: I do not understand what the thymine ADPr is blocking in the model.

6. Page 7, line 32: is it Extended Data Fig.3"c"? I cannot find the description of the results presented in Extended Data Fig.3d

7. I agree on the lack of significance difference between TRF1-DarT and TRF1-DarTmut in wild-type cells (Fig.4a). However, the representative image shows otherwise. Would it be possible to make it more in line with the data?

Reviewer #3 (Remarks to the Author):

The authors have responded satisfactorily to my comments. Nice study!

Author Rebuttal, first revision:

I have few suggestions:

1. Extended Data Fig.1a: is + PARGi? Also PARG is not really introduced in the text.

Response: Yes. This is indicated in the figure and figure legend. The introduction now includes this sentence: PARP activity is counteracted by ADP-ribosylhydrolases, including PARG (Poly-ADP-ribose Glycohydrolase), the major enzyme that degrades PAR chains and other mono-ADPr specific enzymes like ARH3 (ADP-ribosylhydrolase 3).

2. There are still some points in the text where POT1 is indicated as regulating specifically the "lagging" strand replication (Page 5, line 16, 19), when POT1 regulates the overhang metabolism after replication on both strands.

Response: We had referred to it in the context of the ADPr, which appears selectively on the lagging strand. However, we have amended any potentially misleading/confusing statements. Hope that is ok.

3. POT1-F62A does not seem to be completely "unable to" suppress the DNA-ADPr formation (Page 6, line 5 and Extended data Fig.2h). Which is the p-value compared to POT1 KO?

Response: We amended the figure and text to state "However, the expression of POT1-F62A, a POT1 ssDNA binding mutant whose expression hyperextends the overhang⁴² (Extended Data Fig. 2g,h) did not suppress DNA-ADPr as efficiently (Extended Data Fig. 2e,f)."

The p values are:

POT1 KO vs F62A: 0.0042 **
 POT1 KO vs WT: <0.0001 ****
 POT1 WT vs F62A: 0.0116 *

4. Extended data Fig.2h: shouldn't the p-value be shown compared to TARG1 siRNA instead than NT siRNA?

Response: We amended the figure. Thank you!

5. Figure 3a: I do not understand what the thymine ADPr is blocking in the model.

Response: We are unsure what is referred to here. The figure does not show the thymidine ADPr blocking anything. Perhaps the reviewer is referring to the arrows? To clarify, we added text describing what actions the arrows signify.

6. Page 7, line 32: is it Extended Data Fig.3”c”? I cannot find the description of the results presented in Extended Data Fig.3d.

Response: Apologies. This was inexplicably removed in pdf rendering. This has been corrected and highlighted.

7. I agree on the lack of significance difference between TRF1-DarT and TRF1-DarTmut in wild-type cells (Fig.4a). However, the representative image shows otherwise. Would it be possible to make it more in line with the data?

Response: We investigated this. First, to help allay confusion, we ordered the panels to match the graph, i.e., WT and mut conditions. This helps clarify. We also replaced the panels where necessary to improve the presentation of the data. We believe these images adequately represent the data. Thank you for the suggestion.

Final Decision Letter:

Message: 18th Mar 2024

Dear Dr. O'Sullivan,

We are now happy to accept your revised paper "Deregulated DNA ADP-Ribosylation impairs telomere replication" for publication as an Article in Nature Structural & Molecular Biology.

Your paper will be published online soon after we receive proof corrections and will appear in print in the next available issue. You can find out your date of online publication by contacting the production team shortly after sending your proof corrections.

Please note that *Nature Structural & Molecular Biology* is a Transformative Journal (TJ). Authors may publish their research with us through the traditional subscription access route or make their paper immediately open access through payment of an article-processing charge (APC). Authors will not be required to make a final decision about access to their article until it has been accepted. Find out more about Transformative Journals

Sincerely,

Dimitris Typas
Associate Editor
Nature Structural & Molecular Biology
ORCID: 0000-0002-8737-1319